

# iHydroSlide3D v1.0: an advanced hydrological-geotechnical model for hydrological simulation and three-dimensional landslide prediction

Guoding Chen[1,3], Ke Zhang[1,2,3,4], Sheng Wang[1,2,3], Yi Xia[1,3], and Lijun Chao[1,2,3,5]

[1]State Key Laboratory of Hydrology-Water Resources and Hydraulic Engineering, Hohai University, Nanjing, Jiangsu, 210098, China
[2]Yangtze Institute for Conservation and Development, Hohai University, Nanjing, Jiangsu, 210098, China
[3]College of Hydrology and Water Resources, Hohai University, Nanjing, Jiangsu, 210098, China
[4]CMA-HHU Joint Laboratory for Hydrometeorological Studies, Hohai University, Nanjing, Jiangsu, 210098, China
[5]College of Agricultural Science and Engineering, Hohai University, Nanjing, Jiangsu, 210098, China

*Correspondence to*: Ke Zhang (kzhang@hhu.edu.cn)

**Abstract.** Forecasting flood–landslide cascading disasters in flood- and landslide-prone regions is an important topic within the scientific community. Existing hydrological-geotechnical models mainly employ infinite or static 3D stability model and very few models have incorporated the 3D landslide model into a distributed hydrological model. In this work, we modified a

3D landslide model to account for slope stability under various soil wetness states and then coupled it with the Coupled Routing and Excess STorage (CREST) distributed hydrology model, forming a new modelling system called iHydroSlide3D v1.0. The model features the feasibility of applying flexibly different simulating resolutions for hydrological and slope stability submodules by embedding a soil moisture downscaling method. For a large-scale application, we paralleled the code and elaborated several computational strategies. The model produces a relatively comprehensive and reliable diagnosis for flood-

landslide events, including (i) complete hydrological components (*e.g.*, soil moisture and streamflow), (ii) a landslide susceptibility assessment (factor of safety and probability of occurrence), and (iii) a landslide hazard analysis (geometric properties of potential failures). We evaluated the plausibility of the model by testing it in a large and complex geographical area, the Yuehe River Basin, China, where we attempted to reproduce cascading flood–landslide events. The results are well verified at both hydrological and geotechnical levels. iHydroSlide3D v1.0 is therefore appropriately used as an innovative tool

for assessing and predicting cascading flood–landslide events once the model is well calibrated.

## 1 Introduction

Landslides represent mass-movement processes in hilly and mountainous environments and pose significant threats to human lives and properties (Hong et al., 2006;He et al., 2016). Rainfall events characterized by short-duration but high-intensity precipitation can substantially change the soil state of unlithified soil mantle or regolith (Srivastava and Yeh, 1991;Iverson,

2000;Baum et al., 2010), and thus affect hillslope stability and cause flash floods in channels. Forecasting flood–landslide





hazards and correspondingly evacuating people from hazardous zones in advance are widely regarded as a critical risk reduction strategy (Abraham et al., 2021). However, to date, it is still challenging to accurately and reasonably forecast the landslides due to the complex natural processes and the interweaving hydrological, geomorphic, and geotechnical mechanisms (Sidle and Bogaard, 2016;Guzzetti, 2021).

35        Modelling of landslide susceptibility can be appropriately accomplished by adopting a variety of approaches, including statistical methods (Guzzetti et al., 2007;Segoni et al., 2018), physically-based models (Baum et al., 2010;He et al., 2016;Zhang et al., 2016), and geotechnical approaches (van Westen et al., 2006) among others. Among them, the deterministic and physically-based models (PBMs) are popularly used for modelling the spatiotemporal susceptibility of landslides. Some of these approaches attempt to define a direct correlation between rainfall depth and slope stability under some simplified

hypotheses (Montrasio and Valentino, 2008;Liao et al., 2010). These models are useful for regional landslide stability assessment but fail to reproduce cascading flood–landslide disasters in catchments. More recently, efforts have been devoted to coupling the sound hydrological models with more or less complex landslide models (Baum and Godt, 2010;Lepore et al., 2013;He et al., 2016;Zhang et al., 2016;Aristizábal et al., 2016;Wang et al., 2020). Such hydrological-geotechnical models include physical representations of precipitation, evapotranspiration, infiltration with continuous soil moisture accounting,

runoff routing, and the slope stability module. However, most of them rely on infinite slope stability models (*i.e.*, one-dimensional models), which are based on the assumption of planar shallow failures and fail to capture the complexity of landslide geometry in many landscapes where shallow- and deep-seated landslides inherently coexist (Zêzere et al., 2005;Mergili et al., 2014b;Tran et al., 2018). To this end, three-dimensional slope stability models (3D models) are proposed to cope with more complex scenarios (Mergili et al., 2014a;Reid et al., 2015).

Until now, as reviewed by Vandromme et al. (2020), the existing hazard software for the implementation of spatial PBMs mainly employs the one-dimensional (1D) or two-dimensional (2D) methods for slope stability calculation. The 3D approaches like Scoops3D (Reid et al., 2015) and r.slope.stability (Mergili et al., 2014a) are only practical for static conditions such as imposed water level and fully saturated soil state. Researchers have attempted to combine the hydrological part of the Transient Rainfall Infiltration and Grid-Based Regional Slope-Stability (TRIGRS, a well-known, publicly available software)

model (Baum et al., 2010) with a 3D model and analyzed the hillslope stability on a regional scale (Tran et al., 2018;He et al., 2021). As a matter of fact, to the best of our knowledge, there are still very few fully coupled hydrological-geotechnical models that are capable of performing in a large scale and producing 3D information of landslide disasters. The progress is hindered by complicated model structures and considerable computational loads. The latter is inevitable and is an inherent feature for PBMs when the applications are conducted at a large scale using the 3D models (Zieher et al., 2017). Another problem that

will be involved is the selection of computational spatial resolution. Hydrological modelling with a coarse spatial resolution (*e.g.*, 1 km resolution or coarser) but a large-scale coverage has been widely available with the increasing availability of meteorological and land surface data (Xue et al., 2013a;Chao et al., 2019;Chao et al., 2021). However, such a resolution is insufficient to capture the slope failures on hillslope scales, particularly for the landslide events that usually occur within an area of only tens or hundreds of squared meters (Chen et al., 2017). Moreover, it is not wise to unlimitedly refine the mesh





resolution of the hydrological model over a relatively large region. A strategy to tackle the differential needs for computational resolutions among the submodules is essential (Wang et al., 2020).

A comprehensive assessment for landslide disasters is generally composed of three parts (Vandromme et al., 2020): a landslide inventory, a landslide susceptibility analysis (usually denotes factor of safety ($FS$) and probability of occurrence), and a landslide hazard analysis (*i.e.*, magnitude that takes into account the area and volume of failure). Among them, the

landslide hazard analysis is not very common as the ordinary 1D models cannot represent the geometric properties of landslides. Previous studies for this purpose are more inclined to use available landslide datasets (Guzzetti et al., 2009;Brunetti et al., 2009;Klar et al., 2011) and advanced sensing and photogrammetric methods and techniques (*e.g.*, aerial photograph interpretation, high-resolution imagery, and LiDAR interpretation) (Lacroix, 2016). However, in many cases, the landslide data are not well documented or insufficient data is unfavourable to support such analysis (*e.g.*, only failure locations are

recorded). Performing the landslide hazard analysis in such cases is necessary but difficult to implement.

In this work, we developed an innovative physical-based integrated hydrological processes and 3D slope stability modelling framework, which is called the **i**ntegrated **Hydro**logical processes and **3**-**D**imensional land**Slide** prediction model (iHydroSlide3D v1.0), by coupling a distributed hydrological model with a newly-developed 3D geotechnical model. To alleviate the chronic contradiction of mesh resolutions required for hydrological and landslide simulations, we adopted the soil

downscaling method to handle the soil moisture. The iHydroSlide3D v1.0 is built on a parallel computational design, allowing the code to run efficiently on a multi-core machine. The code was tested in a large and complex geographical area, the Yuehe River Basin of western China, where we attempted to reproduce cascading flood–landslide events.

The paper is organized as follows. We first describe the basic theories of submodules and main features of the framework in Section 2. In addition, we also elaborate the strategies for model implementation in Section 2. In Section 3, we

introduce a case study and associated materials required for model simulation and evaluation. Results are presented in Section 4, which are mainly focused on the evolution processes of a historical storm trigerred cascading flood-landslide events. Finally, we discuss the results and summarize the conclusions in Section 5.

## 2 The integrated hydrological-geotechnical model framework: iHydroSlide3D v1.0

### 2.1 Overall structure

iHydroSlide3D v1.0 is a physical-based modelling framework that accounts for both hydrological and geotechnical processes. The model mainly includes the following modules: (i) a distributed hydrological model based on the Coupled Routing and Excess Storage (CREST) model, (ii) a newly developed 3D landslide model, and (iii) a soil moisture downscaling method. The model can currently process two sets of data with different resolutions, allowing to simultaneously modelling hydrological and geotechnical processes with different spatial resolutions. iHydroSlide3D v1.0 is coded in MATLAB and is capable of

running in a parallel manner, currently supported by the Linux and Windows operating systems. Detailed descriptions of the model are presented as follows.



## 2.2 Hydrological model: the Coupled Routing and Excess STorage Model

A physical-based hydrological model, *i.e.*, the Coupled Routing and Excess STorage (CREST) (Wang et al., 2011;Khan et al., 2011;Shen et al., 2016;Xue et al., 2013b) is adopted to simulate hydrological processes that trigger the rainstorm-induced landslide events. The CREST model was first developed by University of Oklahoma (http://hydro.ou.edu) and NASA SERVIR Project Team (www.servir.net) and served for predictions of flash floods caused by rainfalls on its early-version stage (Wang et al., 2011). The model is further enhanced by considering the Multi-Radar Multi-Sensor (MRMS) forcing data and has been used for hydroclimatology studies such as extreme hydrological events (*e.g.*, floods and droughts) (Zhang et al., 2015;Khan et al., 2011) and statistical and hydrological evaluation in ungauged basins (Xue et al., 2013a). The CREST is run in a distributed fashion via a cell-to-cell design concept, while the coupling between overland flow generation and routing scheme allows a realistic and detailed simulation of hydrological variables such as soil moisture, which plays a major role in determining the stability of a slope. More recently, several coupled hydrological-geotechnical models based on the CREST model such as CRESLIDE (He et al., 2016) and iCRESTRIGRS (Zhang et al., 2016) have emerged as the application evolves. These models, counting on the hydrological simulation of the CREST, have achieved their capability of back-calculation and/or prediction for rainfall-triggered landslides. As a consequence, CREST has been comprehensively and extensively evaluated regarding its hydrological simulation skill and its flexibility for coupling. A detailed description of the CREST can be found in Wang et al. (2011) and Xue et al. (2015). For better understanding the work of this study, it is still important to briefly review the principal theories of the CREST model here.

The CREST is driven by precipitation and potential or actual evapotranspiration. The rainfall-runoff generation processes are computed at each cell, starting with accounting for its received precipitation at each time step ($P$). After $P$ passes the canopy layer and deducts canopy interception, the excess precipitation ($P_{\text{soil}}$) then reaches the soil surface. A conceptual variable infiltration curve (VIC), originated from the Xinanjiang Model (Zhao, 1992) and later adopted by the VIC model (Liang et al., 1994), is used to further divide the $P_{\text{soil}}$ into excess rain ($R$) and infiltration water ($I$). The CREST assumes that each soil column is capable to store a maximum water depth, which is regarded as the infiltration capacity ($i$) and varies over an area in the following relationship:

$$i = i_m \left[ 1 - (1-a)^{\frac{1}{b}} \right], \tag{1}$$

where the $i_m$ is the maximum infiltration capacity of a cell and strongly depends on the soil properties; $a$ is a fraction number of a grid cell and $b$ is an empirical shape parameter. Under this assumption, the amount of water available for excess rain ($R$) and infiltration ($I$) can be further expressed as:

$$I = \begin{cases} W_{\text{m}} - W, i + P_{\text{soil}} \geq i_{\text{m}} \\ (W_{\text{m}} - W) + W_{\text{m}} \cdot \left[ 1 - \dfrac{i + P_{\text{soil}}}{i_{\text{m}}} \right]^{1+b}, i + P_{\text{soil}} < i_{\text{m}} \end{cases}, \tag{2}$$

$$R = P_{\text{soil}} - I, \tag{3}$$





where $W_m$ denotes the maximum water capacity of a cell; $W$ represents the total mean water of the three soil layers. $R$ can be further partitioned into overland and subsurface flows by comparing $P_{\text{soil}}$ to the infiltration rate of the first layer ($K$), which is closely related to the soil saturated hydraulic conductivity ($K_{\text{sat}}$). Then CREST adopts the multi-linear reservoir method to
simulate the cell-to-cell routing of overland and subsurface runoff at each time step. The model can better take into account the interaction between the surface and subsurface flows by coupling the runoff-generation process and the routing scheme (Wang et al., 2011).

**2.3 3D stability model based on sliding surface**

The 3D slope-stability analysis model was originally derived to describe the characteristics of a potential failure (Hovland,
1979). This model has no iteration procedure but computes the $FS$ directly compared to the slope-stability models established based on Bishop (1955) and Janbu et al. (1956). Embedded in geographic information systems (GIS), the model composes a slope failure with column units, expressed as grid cells in GIS (software like 3DSlopeGIS) (Xie et al., 2003;Xie et al., 2004;Xie et al., 2006). More recently, progress has been made in a more sophisticated software r.slope.stability (Mergili et al., 2014a;Mergili et al., 2014b) that have the capacity to perform on a regional scale via a parallel computational technique. More
importantly, the 3D slope-stability model demonstrates to be effective on both shallow and deep landslides, thus better behaves as a robust geotechnical tool and has a potential for wide applications (Zieher et al., 2017;Palacio Cordoba et al., 2020).

However, to implement on a large scale, the previous versions of the 3D stability model treat the hydrological component (*e.g.*, transient soil moisture and water level) as static or imposed inputs, failing to consider the time-dependent hydrological processes (Mergili et al., 2014b;Mergili et al., 2014a). In this work, the model is extended to take into account
spatiotemporal variations of water fluxes and storages on regular grids by introducing the hydrological module. Following an assumption of being ellipsoidal or truncated in shape, the potential slope failures are randomly generated over a whole study region. When applied in a regional assessment, the theory of the model can be mainly divided into the following two parts.

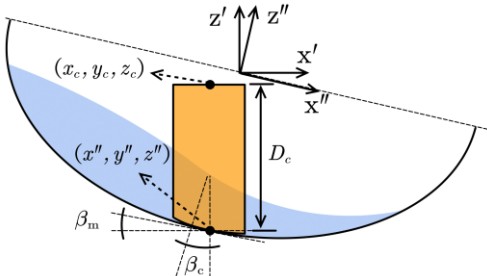

**Figure 1: Coordinate systems involved in an arbitrary ellipsoid.**

**2.3.1 Coordinate transformation and geometric derivation**

Three levels of the coordinate system involved in this model are (i) GIS coordinate system ($x,y,z$) over the whole study area (Fig. 1), (ii) Cartesian coordinate ($x',y',z'$) of each potential failure, and (iii) ellipsoid coordinate system ($x'',y'',z''$) along the



direction of the steepest slope in a single ellipsoid. The center of each ellipsoid $(x_c, y_c, z_c)$ is randomly generated within the study area, while the GIS coordinate system is simultaneously transformed to the Cartesian coordinate from a ground

perspective (Mergili et al., 2014b):

$$x' = (x-x_c)\cos\alpha + (y-y_c)\sin\alpha, \tag{4}$$

$$y' = (y-y_c)\cos\alpha - (x-x_c)\sin\alpha, \tag{5}$$

where $\alpha$ is the main dip direction of the ellipsoid; $x''$ is easily derived as $x'' = \frac{x'}{\cos\beta}$ ($\beta$ is the main inclination of the ellipsoid, see in Fig. 2); $y''$ is identical to the $y'$ axis; $z'$ is identical to the $z$ axis (Fig. 1). Then we need to filter the grid cells

encompassed by this random ellipsoid, meeting the following condition:

$$\frac{x'}{a_e^2} + \frac{y'}{b_e^2} \leqslant 1, \tag{6}$$

where $a_e$ and $b_e$ are half axes of the ellipsoid, following the $x''$ and $y''$ axes, respectively. These geometric lengths are randomly generated within user-defined ranges. To facilitate the derivation, we give a value of another half axes of the ellipsoid ($c_e$) beforehand, which, in fact, is highly dependent on failure depth and should be reconsidered in following sections. Hence,

with regard to an ideal ellipsoid, the above variables need to satisfy the basic equation of the ellipsoid:

$$\frac{(x'' + \Delta x'')^2}{a_e^2} + \frac{y''^2}{b_e^2} + \frac{\Delta x'^2}{c_e^2(\tan\beta)^2} = 1. \tag{7}$$

By solving the intermediate variable $\Delta x''$, the $z''$ can be computed as:

$$z'' = \frac{\Delta x''}{\tan\beta}. \tag{8}$$

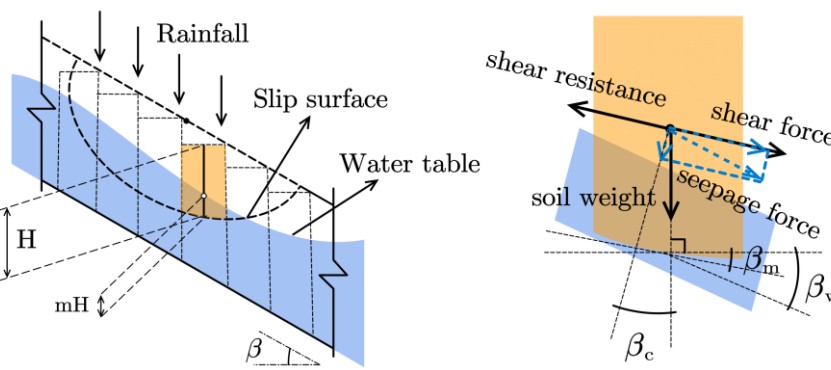

**Figure 2: Typical longitudinal section of an ellipsoid used as slip surface in iHydroSlide3D v1.0: (a) overall features involved in a potential failure, and (b) forces acting at each column considering the groundwater effect.**

Finally, we transformed it back into the GIS coordinate system:

$$z_{slip} = z_c + \frac{(z'' - x'\sin\beta)}{\cos\beta}, \tag{9}$$





where $z_\text{slip}$ is the elevation of the considered cell in the ellipsoid. Hereto we get all coordinates once a random ellipsoid is

generated. We further note that such procedure is required for each random ellipsoid (*i.e.*, each random loop) and thus is time-consuming particularly in a regional map system. The countermeasures will be introduced in the following sections.

**2.3.2 Basic hydrogeological mechanics**

This study adopted a conceptual parameter $m$ to better simulate the soil moisture of each considered column in a random ellipsoid (see in Fig. 2). The parameter originated from Montrasio and Valentino (2008) and were later represented in further

applications (Liao et al., 2010;He et al., 2016). The parameter $m$ is a distributed value ranging from 0 to 1 and is controlled by hydrologic mechanisms (Fig. 2), which further impacts the matric suction and results in occurrences of landslides (Baum et al., 2010). More specifically, the apparent cohesion is strongly dependent on matric suction, which in turn is related to the degree of saturation of the soil column ($S_r$) (Montrasio and Valentino, 2008):

$$c_\psi(t) = \delta \cdot S_r \cdot (1 - S_r)^\lambda \cdot (1 - m)^\alpha, \qquad (10)$$

where $\delta$ is a soil-type parameter and mainly refers to the peak shear stress at a failure layer; $\alpha$ and $\lambda$ are numerical parameters to estimate the extreme points of the shear strength curve versus $S_r$ and versus the degree of saturation of the soil, respectively. Then the total cohesion ($C'$) is computed as follow:

$$C' = c' + c_\psi(t), \qquad (11)$$

where $c'$ is effective cohesion depending on soil type and is treated as a constant value associated with each grid cell. The

failures may take place in both partially and fully saturated scenarios (Lu and Likos, 2006;Lu and Godt, 2013); the latter should take the seepage force ($S$) into account (Collins and Znidarcic, 2004). Considering the inter-slice forces in this model, the seepage force is computed according to the hydraulic gradient, reflecting a more general situation in the hillslope (King, 1989;Mergili et al., 2014b). Note that the seepage force is only considered in soil columns satisfying $m > 0$. Besides, the grid cell that has a low elevation is excluded from the considered ellipsoid by comparing $z_\text{slip}$ and $z_c$:

$$D_c = z_c - z_\text{slip}. \qquad (12)$$

For the soil column satisfying both of the conditions: $m > 0$ and $D_c > 0$, the seepage force can be approximated by the slope ($\beta_w$) and aspect ($\alpha_w$) of the groundwater table (Fig. 2), acting in the direction of the hydraulic gradient (Mergili et al., 2014b;Mergili et al., 2014a):

$$S = \gamma_w \cdot dx \cdot dy \cdot mH \cdot \sin\beta_w, \qquad (13)$$

where $\gamma_w$ is the specific weight of water; $dx$ and $dy$ are the cell size, depending on the resolution of input data. To further transfer the seepage force from hydraulic gradient to sliding direction, $S$ is first divided into horizontal ($S_h$) and vertical ($S_v$) components (Fig. 2):

$$S_h = S\cos\beta_w \text{ and } S_v = S\sin\beta_w. \qquad (14)$$

$S_v$ is irrelevant to the direction, while $S_h$ needs to be further projected according to the dip direction of grid column ($\alpha_c$) and

the main inclination direction of the slip surface given by:





$$S_{ch} = S_h \cos(\alpha_w - \alpha_c) \text{ and } S_{mh} = S_h \cos(\alpha_w - \alpha). \tag{15}$$

Conforming to the orthogonality rule, the projected seepage force $(S_c, S_m)$ and their vertical angle $(\beta_{S_c}, \beta_{S_m})$ can be expressed as:

$$
\begin{aligned}
S_c &= \sqrt{S_v^2 + S_{ch}^2}; S_m = \sqrt{S_v^2 + S_{mh}^2} \\
\cos \beta_{Sc} &= \frac{S_{ch}}{S_c}; \ \beta_{Sm} = \frac{S_{mh}}{S_m}
\end{aligned}
. \tag{16}
$$

The final expression of the seepage force acting on each grid column can be written as normal and slope-parallel components:

$$N_s = S_c \sin(\beta_{Sc} - \beta_c); \ T_s = S_m \cos(\beta_{Sm} - \beta_m). \tag{17}$$

The soil weight $(G')$, considering the variant degree of saturation and under the condition of $D_c > 0$, is derived as:

$$G' = dx \cdot dy \cdot [\gamma_d D_c + \gamma_w \cdot mH \cdot (n-1) + \gamma_w (D_c - mH) n S_r], \tag{18}$$

where $\gamma_d$ is the unit weight of the dry soil; $n$ and $S_r$ represent the porosity and soil saturation degree, respectively. Based on

the limited equilibrium condition, the model assesses the critical scenarios by calculating the $FS$, which can be mechanically subject to the stabilizing and destabilizing actions. Summarizing the derivations above, the extended version of the 3D slope-stability equation can be written as follow:

$$FS = \frac{\sum_c [(C' + \delta \cdot S_r \cdot (1 - S_r)^\lambda \cdot (1 - m)^\alpha) \cdot A + (G' \cos \beta_c + N_s) \tan \varphi] \cos \beta_m}{\sum_c (G' \sin \beta_m + T_s) \cos \beta_m}, \tag{19}$$

where $\varphi$ is the friction angle; $\beta_c$ and $\beta_m$ denote the dip and apparent dip of the slip surface at a considered soil column,

respectively; $A$ is the slip surface area of each column and can be computed as:

$$A = dx \cdot dy \frac{\sqrt{1 - (\sin \beta_{xz})^2 (\sin \beta_{yz})^2}}{\cos \beta_{xz} \cos \beta_{yz}}, \tag{20}$$

where $\beta_{xz}$ and $\beta_{yz}$ are apparent dips of x- and y-axis, respectively. The relationships between the apparent dips and main sliding direction assigned to each soil column can be expressed as (Xie et al., 2003):

$$
\begin{aligned}
\tan \beta_m &= \tan \beta_c |\cos(\alpha_c - \alpha)| \\
\tan \beta_{xz} &= \tan \beta_c \sin \alpha_c \\
\tan \beta_{yz} &= \tan \beta_c \cos \alpha_c
\end{aligned}
. \tag{21}
$$

The model diagnoses whether the landslide is stable or not by comparing the value of $FS$ with a critical value that usually set to 1. At the same time, for each random ellipsoid, the volume and area of a failure can be approximated by:

$$V_L = \sum D_c \cdot dx \cdot dy, \tag{22}$$

$$A_L = \sum dx \cdot dy. \tag{23}$$

It is worth noting that the model can serve in a stand-alone manner by directly imposing soil moisture and groundwater table. However, in a more practical sense, the landslide model is coupled with the hydrological model.


### 2.4 Soil moisture downscaling method

A near-conservative downscaling method of soil moisture (Droesen, 2016;Wang et al., 2020) is adopted here to link different-resolution-based submodules in the iHydroSlide3D v1.0, *i.e.*, the relatively coarse-resolution hydrological model and the fine-resolution 3D slope-stability model. The method relates the soil moisture with the topographic wetness index (TWI) by proposing a conversion parameter, the wetness coefficient ($K_w$). Readers may refer to Wang et al. (2020) for more detailed descriptions. This method helps the hydrological module produce soil moisture with a higher resolution that can be seamlessly utilized by the landslide module. The method has demonstrated its effectiveness (Wang et al., 2020) and is necessary for a hydrogeological-type model to balance the tedious computational tasks and accuracy.

### 2.5 Coupling strategy and model implementation

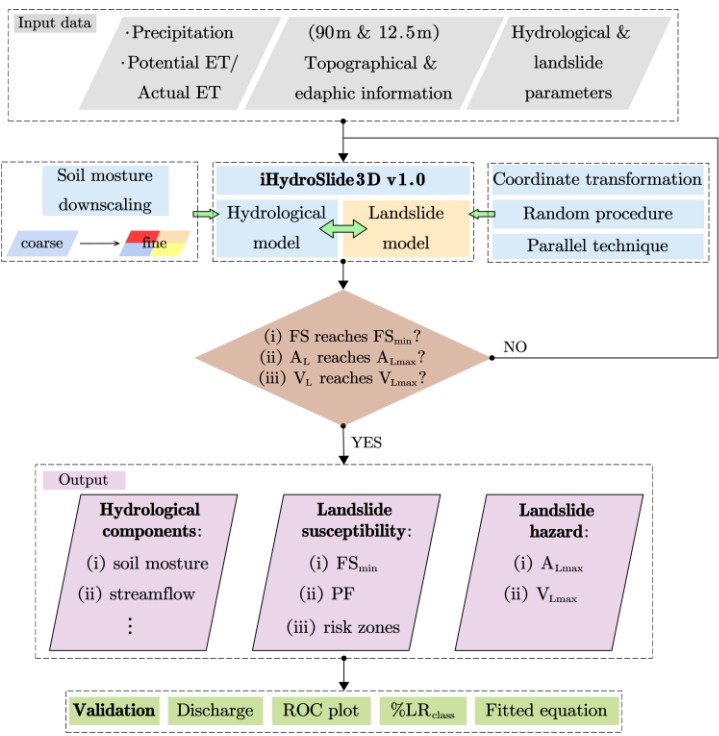

**Figure 3: Flow chart illustrating the work process of the iHydroSlide3D v1.0 model.**

The iHydroSlide3D v1.0 mainly consists of three sub-modules: (i) hydrological model CREST, (ii) soil moisture downscaling method, and (iii) 3D landslide-stability model (Fig. 3). The CREST undertakes the complete computational tasks of hydrologic processes, including interception by vegetation, water infiltration, runoff generation, cell-to-cell routing, and re-infiltration on each grid cell in the course of excess surface runoff moving from upstream to downstream, of which the infiltration and re-infiltration play the most important role on the coupled hydrology-slope stability processes. The landslide model inherits the hydrological variables from the hydrological model and acts as a slope-stability monitor. The complete simulation cycle is





seamlessly facilitated by the downscaling module. To elucidate the implementation of the iHydroSlide3D v1.0 model, we

present the logical framework in Fig. 3 and summarize the detailed coupling strategy in the following aspects:

1. Instead of directly linking the soil moisture with rainfall intensity, the model takes the water loss into account due to the interception and evapotranspiration. The hydrological module helps to better simulate antecedent conditions such as soil moisture and cumulative infiltration. As a consequence, the parameter $m$ is updated as a spatiotemporal variable ($m_t$) (He et al., 2016):

$$m_t = \frac{W_t}{nD_t(1 - S_r)},\tag{24}$$

where $W_t$ is the mean water amount of the three soil layers on a given grid cell. $S_r$ can be computed as:

$$S_r = \frac{W_t}{W_m}.\tag{25}$$

$D_t$ is the landslide's initiation depth for various soil states and is largely impacted by soil heterogeneity and hydraulic properties (Lu and Godt, 2008). Therefore, $D_t$ is determined by infiltration processes at time $t$ (He et al., 2016):

$$D_t = \sqrt{\frac{2K_sH_ct}{\theta_n - \theta_0}},\tag{26}$$

where $K_s$ is saturated hydraulic conductivity; $H_c$ is capillary pressure; $\theta_n$ is volumetric water content of the saturated soil; $\theta_0$ is initial water content of the soil. Note that the $m_t$, $S_r$, and $D_t$ are gridded values.

2. We prepare two sets of data with different resolutions: a relatively coarser hydrological resolution and a finer landslide resolution. Once the soil moisture is calculated for all coarser grid cells, the soil moisture downscaling module is

activated to calculate a new soil moisture map in a finer resolution to fit the spatial resolution of the landslide model ($SM_{\mathrm{Hydro}} \rightarrow SM_{\mathrm{Land}}$).

3. In each simulation time step, the model generates a large number of ellipsoidal tested landslides with random geometric center and ellipsoid length and width. The latter is constrained by the range of maximum and minimum values, which are determined from field investigation and regarded as the input parameters. Each random ellipsoid adopts maximum

soil depth as another geometric length ($c_e$) among the encompassed cells ($D_t = \max\{D_{cell_1}, D_{cell_2}, D_{cell_3}, \cdots\}$). The coordinate transformation and related geometric derivation are then tackled according to Sect. 2.3.1. Next, each tested landslide slip surface corresponds to a $FS$ value, based on the mechanical analysis described in Sect. 2.3.2.

4. Attributable to random strategy in the model architecture, any tested landslide will be possibly overlapped by another one, resulting in the confusing values of $FS$ for each considered grid cell. In other words, each grid cell has a chance

to be stable or unstable. For instance, as illustrated in Fig. 4, grid cell #a is estimated to be unstable in a tested landslide #3 but stable in the tested landslides #4 and #5. In this work, we assign the minimum value of $FS$ ($FS_{\min}$, Fig. 4b) and failure probability ($PF$, Fig. 4a) to each grid cell (Mergili et al., 2014b):

$$FS_t = \min\{FS_{L_1}, FS_{L_2}, FS_{L_3}, \cdots\},\tag{27}$$





$$PF_t = \frac{\sum PF_{FS<1}}{\sum PF_{FS<1} + \sum PF_{FS>1}}. \tag{28}$$

The model counts all possible values of $FS$ and, based on a sufficiently large number of ellipsoids and possible ellipsoid dimensions, determine the final values of $FS$ and $PF$ for each considered grid cell. Similarly, each grid cell belongs to a maximum value of volume and area of a failure:

$$V_{Lmax} = \max\{V_{L_1}, V_{L_2}, V_{L_3}, \cdots\}, \tag{29}$$

$$A_{Lmax} = \max\{A_{L_1}, A_{L_2}, A_{L_3}, \cdots\}. \tag{30}$$

The records of these values are only effective in the current simulation moment and will be reset as the simulation time moves forward. As the hydrological process evolves, the model is able to dynamically assess the slope stability and treats the slope-stability assessment indices as variables.

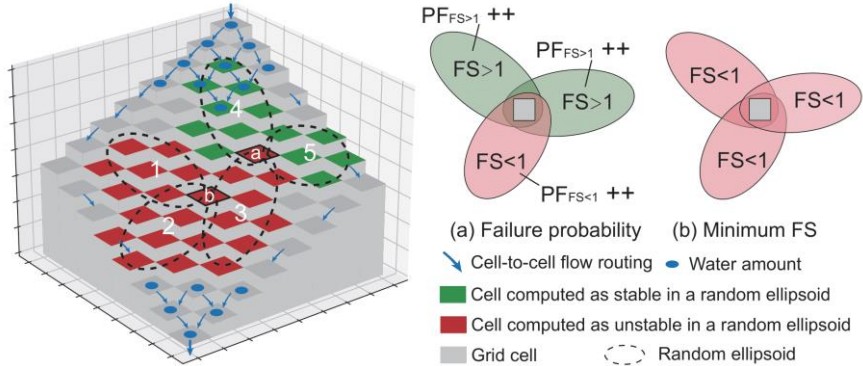

**Figure 4: Cell-to-cell routing scheme and potential landslides generated across the grid in the iHydroSlide3D v1.0 model: (a) and (b)**
**illustrate the definitions of $PF$ and $FS$ within the framework, respectively.**

We believe that the above variables will reach the computational convergence provided the number of tested ellipsoid is sufficient enough. As a requirement, the "density" of ellipsoids is recommended to reflect the total number over the study area (Mergili et al., 2014b):

$$d_s = n\frac{A_p}{A_s} = n\frac{\pi(a_{e|max} + a_{e|min})(b_{e|max} + b_{e|min})c_t}{16A_s}, \tag{31}$$

where $n$ is the chosen total number of tested landslide; $A_p$ is average vertical projection of area of a single tested landslide; $A_s$ is the extent of the study area; $a_{e|max}$, $a_{e|min}$, $b_{e|max}$, and $b_{e|min}$ are the upper and lower limits for randomization of ellipsoid length and width; $c_t$ is a dimensionless correction factor and is set to the average cosine of the slope (Mergili et al., 2014a). Note that the $d_s$ is strongly related to constraints of the random length and width and resolution of the digital elevation model, for which should be tested and set to an appropriate value before meaningful application. We also acknowledge that the model
outcomes the worst-case situation ($FS_{min}$, $V_{max}$, and $A_{max}$), however, along with the probability of the failure ($PF$).





## 2.6 Auxiliary computational strategy

There are two main computational bottlenecks in the model, which causes a large computational burden: (i) the operation of coordination transformation described in Sect. 2.3.1 is required for each random ellipsoid and, even in a single simulation time, will be executed $n$ times (see in Eq. 31); (ii) the 3D slope stability model is inherently complicated and is also repeatedly

calculated for $n$ times, leading to tedious computational tasks. To cope with the above computation-intensive problems, the following strategies are adopted in this work:

  1. We use the smallest and variable "moving window" to just encompass a single ellipsoid being tested. Each ellipsoid can correspond to a small coordinate matrix, in which the coordination transform occurs, to avoid computing the entire study area.

2. iHydroSlide3D v1.0 is built upon a parallel computing framework and has a capacity of running on multicore processors or computer clusters. The model also provides the option to call the local maximum or a user-defined number of cores up to the limit of the hardware. The model divides the study area into user-defined number of tiles and each of them is processed independently in parallel. All computing tasks need to be queued until there are free computing cores. The slope-stability information is computed and counted for each tile and is stored in the computer memory. At the end of each simulation

time step, the model combines all tiles and recalculates the overlapping part of the margin of each "moving window", and then outputs the final results. The model clears the computer memory after the procedure and repeats the above operations in the next simulation period.

## 2.7 Model validation

iHydroSlide3D v1.0 can be mainly evaluated on the hydrological and landslide event levels (Fig. 3). Streamflow observations

from the local gauge stations are utilized for validation of the modeled discharge. The statistical metrics such as Nash–Sutcliffe coefficient of efficiency (NSCE), Pearson correlation coefficient (CC), and relative bias are computed to measure the model performance. Furthermore, more than a single gauge station is necessary when the very large scale or multiple basins are involved. We also expect that the hydrologic process can be further calibrated by soil moisture data if the measurements are available, since soil moisture is more related to slope stability and thus is recommended (Lepore et al., 2013). To validate the

model's predicative capability for landslides, *in situ* measurements (*e.g.*, $L, W, V$, and $A$ of failures) will be ideal data for model validation and refinement. Such data not only serve for evaluation but also provide more hints for the constraint of random procedure and model preparation. However, in most cases, only point-like landslides are available for assessing the performance of initiation location prediction. Two existing synthetic indices $\%LR_{class}$ (Park et al., 2013;Tran et al., 2018) and Receiver Operating Characteristic (ROC) curve (Fawcett, 2006) are used for measure the model performance. Lack of the

specific time for all landslide occurrences, we evaluated the model performance in the worst case of the hydrological conditions. In another word, we would consider a successful prediction if the recorded landslide sites were estimated as failures during the complete rainfall event.





## 2.8 Model inputs and outputs

The input data includes precipitation and evapotranspiration, digital elevation models (DEM), and soil texture and land cover

maps, while the observed river streamflow and the inventory of landslide events are used to calibrate and validate the model. Additional topographic information such as slope angle and direction, which are also needed for the landslide model, are directly derived from the DEM data. Several hydrological parameters like $W_m$ should be carefully prepared before the simulation and will be displayed in the following section. The output variables include all typical hydrological components (*e.g.*, overland runoff, soil moisture, and infiltration information) and landslide assessments ($FS, PF, V_L,$ and $A_L$). Note that

model output is controlled by a user-defined "GlobalControlFile" and the components are thus selected based on the interest of the user. The model calls for two sets of topographic data (see in Sect. 2.5) and all gridded data are either downscaled or aggregated to an objective spatial resolution to ensure the forcing and auxiliary data matching with each other. iHydroSlide3D v1.0 currently supports several different options for file formats (ASCII, TIFF, and TXT) and map projections, of which the Geographic Tagged Image File Format (GeoTIFF) is preferred for its distinct advantage of containing native compression

capabilities, making the file sizes smaller.

## 3 Materials and model setup

We test the iHydroSlide3D v1.0 code in the Yuehe River Basin, Shaanxi Province, China (Fig. 5). The basin has an elevation between 270 to 2700 m a.s.l. and covers a total area of 1100 km². The terrain in this basin is characterized by steep hills, gullies, and valleys, while its flood season is usually accompanied by heavy and frequent rainfall. As a result, this basin is

highly susceptible to slope instability and sliding (Zhang et al., 2019;Wang et al., 2020). In this area, 54 slope failure locations were reported during a rainstorm from July 3th to 4th in 2012 (have no more specific time record). In addition, the discharge of the flash flood was also observed at the outlet of the basin.

Hourly precipitation data were provided by China Meteorological Administration (CMA) based on the observations of gauge stations and were interpolated into a spatial resolution of 3 arc sec (~ 90 m). The potential evapotranspiration (PET)

data were derived from Global Land Data Assimilation System (GLDAS). The 3-h, 0.25° PET data were first downscaled to a resolution of 3" using bilinear interpolation and further downscaled to an hourly scale using linear interpolation. Two different resolutions of DEM (90 m and 12.5 m) from the NASA Shuttle Radar Topography Mission (SRTM) Version 3.0 (SRTM3) DEM and Advanced Land Observing Satellite (ALOS) DEM are used for hydrological and landslide modelling (introduced in Sect. 2.5), respectively. The flow direction (FDR) and flow accumulation maps (FAC) are necessary for hydrological

simulation and can be derived from the DEM map. The slope angle map is optional for hydrological modelling but required for landslide modelling, which can be directly computed through a built-in slope angle calculation function in iHydroSlide3D v1.0. The TWI data were derived using the ESRI ArcGIS and its ArcHydro toolbox. The land cover data were derived from the 30m GlobeLand30-2010 data (Chen et al., 2015). Soil texture was classified into the 12 United States Department of





Agriculture (USDA) soil texture types from the Harmonized World Soil Database (HWSD v1.2) (Wieder et al., 2014) based
on a lookup table (Table 1) shared by both hydrological and landslide modules.

The parameters used for this model are largely related to *a priori* map of soil information and have been generated
by Wang et al. (2020) and Zhang et al. (2016). $W_m$ corresponds to available water capacity between field capacity and wilting
point (Table 1) and is distributed according to both topography and soil texture (Yao et al., 2012;Wang et al., 2020). Saturated
hydraulic conductivity ($k_s$) strongly depends on the soil type and is determined through the pedotransfer look-up table (Table
1). Impervious surface area (ISA) can obviously affect the hydrological process such as infiltration and runoff generation and
is calculated for each grid cell by considering the fractions of artificial surface and wetland in land cover map. For the landslide
module, the constraints of the random landslides are regarded as priori parameters depending on the inventory. The total tiles
divided from the entire area, along with the landslide density and user-defined number of cores, are summarized as related to
parallel computational parameters. All about the basic materials and parameters are briefly listed in Tables 2 and 3.

We run the model on the High-Performance (HP) cluster with 1 manage node and 8 computational nodes (Intel(R)
Xeon(R) CPU E5-2660 v4 @2.00GHz). Each node operates a CentOS with 28 cores and 64GB RAM and reaches a total of
56 threads based on the hyper-threading technology.

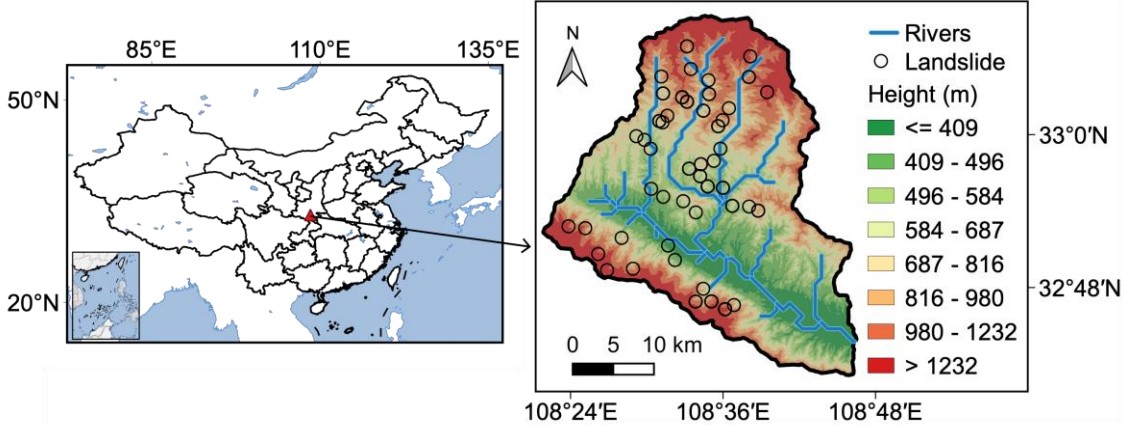

**Figure 5: Locations of the Yuehe River Basin with its elevation and the reported landslide events.**


**Table 1: Lookup table of key parameters for different soil types used in this study (refer to Wang et al. (2020) and Zhang et al. (2016)).**

| USDA Soil Type | Soil Cohesion (kPa) | Saturated Hydraulic Conductivity (m/s) | Porosity | Friction Angle (degree) | Soil Dry Unit Weight (kN/m³) | Field Capacity (m³/m³) | Wilting Point (m³/m³) |
|---|---|---|---|---|---|---|---|
| Silty clay | 30 | $1.06 \times 10^{-6}$ | 0.49 | 18.5 | 18 | 0.36 | 0.21 |
| Clay | 40 | $1.31 \times 10^{-6}$ | 0.47 | 16.5 | 19.5 | 0.36 | 0.21 |



| Silty clay loam | 50 | $1.44 \times 10^{-6}$ | 0.48 | 16.5 | 14 | 0.34 | 0.19 |
|---|---|---|---|---|---|---|---|
| Clay loam | 35 | $2.72 \times 10^{-6}$ | 0.46 | 20 | 14 | 0.34 | 0.21 |
| Silt | 9 | $2.05 \times 10^{-6}$ | 0.52 | 26.5 | 16.5 | 0.32 | 0.165 |
| Silt loam | 9 | $2.50 \times 10^{-6}$ | 0.46 | 24 | 14 | 0.3 | 0.15 |
| Sandy clay | 24.5 | $4.31 \times 10^{-6}$ | 0.41 | 22.5 | 18.5 | 0.31 | 0.23 |
| Loam | 10 | $4.53 \times 10^{-6}$ | 0.43 | 22.5 | 13 | 0.26 | 0.12 |
| Sandy clay loam | 29 | $6.59 \times 10^{-6}$ | 0.39 | 20 | 15 | 0.33 | 0.175 |
| Sandy loam | 6 | $1.02 \times 10^{-5}$ | 0.4 | 32 | 15 | 0.23 | 0.1 |
| Loamy sand | 7.5 | $1.78 \times 10^{-5}$ | 0.42 | 28.5 | 20.5 | 0.14 | 0.06 |
| Sand | 5 | $2.44 \times 10^{-5}$ | 0.43 | 40 | 21 | 0.12 | 0.04 |

**Table 2: Description of basic input data used in iHydroSlide3D v1.0.**

| Model input | Description | Value/resolution | Data source |
|---|---|---|---|
| Rain | Precipitation data (mm) | Downscaled to hourly and of 3" resolution | China Meteorological Administration (CMA) based on gauge stations |
| Pet | Potential ET data (mm) | Downscaled to hourly and of 3" resolution | Global Land Data Assimilation System (GLDAS) |
| DEM | Digital elevation model | 90 m and 12.5 m for hydrological and landslide modelling, respectively | SRTM3 DEM (NASA v2.1) and ALOS DEM (Alaska Satellite Facility) |
| FDR | Flow direction | 90 m resolution | Derived from the DEM data using the ESRI ArcGIS ArcHydro toolbox |
| FAC | Flow accumulation | 90 m resolution | Derived from the DEM data using the ESRI ArcGISArcHydro toolbox |
| LANDCOVER | Land surface cover | Aggregated to 90 m resolution | GlobeLand30-2010 (Chen et al., 2015) |
| SOIL | Soil texture map | USDA soil code from 1 to 12 with 90 m resolution | Harmonized World Soil Database (HWSD v1.2, (Wieder et al., 2014)) |





| | | | and the Natural Resources Conservation Service (NRCS) of the US Department of Agriculture |
| --- | --- | --- | --- |
| TWI | Topographic wetness index needed in soil moisture downscaling module | Derived using ESRI ArcGIS and the ArcHydro toolbox based on the slope and the upstream contributing area; Both 90 m and 12.5 m resolution are necessary | NA |


**Table 3: Description of model parameters used in iHydroSlide3D v1.0.**

| Parameters | Description | Value/resolution | Source |
| --- | --- | --- | --- |
| TimeStep | Time step of the simulation (s) | Defined by user | NA |
| ISA | Percentage impervious area (%) | Computed based on land cover map | NA |
| Ksat | Saturated hydraulic conductivity $(mmh^{-1})$ | Derived from soil texture map | NA |
| WM | Available water capacity (mm) | Computed from topography and soil texture | Wang et al. (2020) |
| b | Exponent of the infiltration curve | Determined by soil texture | Flamig et al. (2020) |
| Ncores | Number of parallel computational cores | Defined by user and limited by hardware | NA |
| Landslide Density | Density of the random ellipsoid over the area | Defined in Eq. (31) and chosen as a appropriate after testing | Refer to Mergili et al. (2014a) |
| TotalTile | Number of divisions of study area | Defined by user and should refer to Ncores | NA |
| MAXae | The maximum length of a random ellipsoid (m) | 200 | Landslide inventory |
| MINae | The minimum length of a random ellipsoid (m) | 50 | Landslide inventory |
| MAXbe | The maximum width of a random ellipsoid (m) | 150 | Landslide inventory |
| MINbe | The minimum width of a random ellipsoid (m) | 50 | Landslide inventory |

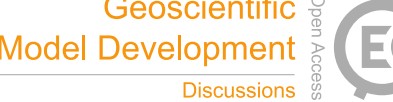

## 4 Results

### 4.1 Evaluation of the soil moisture downscaling method

We first evaluated the impacts and effectiveness of the soil moisture downscaling method, which provides more detailed soil
water information (groundwater) for landslide modelling, and may directly impact the stability assessments. Compared to the
infinite landslide model (Wang et al., 2020), the 3D model can fully consider the grid cells encompassed by an assumed
landslide boundary (elliptical outline, see in Fig. 6). The cells were chosen from the 90-m resolution datasets with different
antecedent soil water amount, of which the single value was converted to a range among over the 7 × 7 map with a 12.5-m
spatial resolution (Fig. 6). The long axis ($a_e$) of the tested ellipse reaches the diagonal of the square as far as possible to
encompass more soil columns, and the potential depth of a failure is set to 2 m. The downscaled soil moisture values are
irregularly distributed (Fig. 6) because they are contributed by several factors with local slope angle as the major one (Wang
et al., 2020). As a consequence, the factor of safety was computed to a different value when using the single or composed soil
moisture values for an assumed landslide (Table 4). In these four test sites, the risks are computed as the worse case situations.
However, in reality, such effects will be more uncertain due to the fact that the location and geometry of a landslide and
associated hydrological conditions are all variable during the modelling. We argue that this downscaling method is necessary
when we perform the iHydroSlide3D v1.0 in a cross-scale manner.

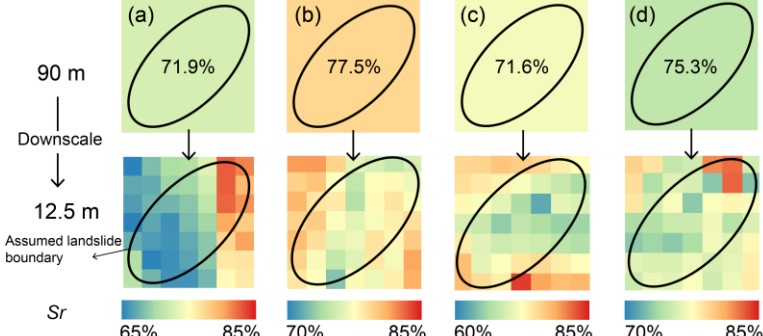

**Figure 6: Soil moisture downscaling results from a coarser resolution (90 m) to a finer resolution (12.5 m). (a)~(d) are four grid cells selected from the 90-m resolution map. The ellipse is the assumed landslide boundary and encompasses the grid cells with the 12.5 m resolution.**

**Table 4: Impacts of soil moisture downscaling on the potential slope failures in terms of the computed FS value.**

| Test cases | Original soil moisture ($m^3/m^3$) | Downscaled soil moisture ($m^3/m^3$) | Original FS | FS from downscaled soil moisture |
|---|---|---|---|---|
| 1 | 71.9% | 66.3%~83.6% | 1.65 | 1.97 |
| 2 | 77.5% | 71.2%~84.5% | 1.45 | 1.81 |
| 3 | 71.6% | 62.6%~86.2% | 1.96 | 2.35 |
| 4 | 75.3% | 68.3%~87.3% | 1.46 | 1.78 |



## 4.2 Testing landslide density

The model requires an appropriate user-defined landslide density that is highly related to model computation efficiency. This value is determined to satisfy the convergence of the results over the study area and, meanwhile, an acceptable level of the running time. Similar work has been done in the previous research (Mergili et al., 2014b) and, equally important, here we further study the relationship between the tile size and random ellipsoid density required. We carried out the convergence tests for three different sizes of a divided tile: $20 \times 20$, $50 \times 50$, and $100 \times 100$ (number of grid cells). For each scenario, we increased the $d_s$ (see in Eq. (31)) value and compared the spatial pattern with the previous $d_s$ step. Two computational targets, the cumulative changed area over the entire region ($\sum$changed pixels area) and cumulative changed $FS$ multiplied by area ($\triangle$ FSR $\times$ area), were used to evaluate the quantity of the convergence results (Mergili et al., 2014b). The total changed pixels area is easier to satisfy the convergence condition, *i.e.* all pixels have been assigned relatively invariant value of $FS$, while another target is strongly affected by the area of the tile (Fig. 7). In general, all the scenarios have similar convergence processes in term of $\sum$changed pixels area (around the 500 in Fig. 7). $\triangle$ FSR $\times$ area is more difficult to converge with the increase of the total area because this cumulative value is closely related the total number of the cells. We note that there exists no theoretical value of landslide density due to the fact that the generation of the potential landslide is totally random. Strictly speaking, $d_s = \infty$ will be an optimum value; however, there will always be a trade-off between the quality and efficiency of the calculation. Further, the increase for overall quality of the prediction cannot be found with a larger adopted density when the $\sum$changed pixels area has converged, which in turn, can significantly increase the computational burden (Mergili et al., 2014a). Besides, the density is mainly determined by constraints for the randomization of ellipsoid dimensions, for which the value would be set based on necessary tests if the model is applied to a new area. For the application in this study area, we consider $d_s = 500$ a sufficiently reasonable approximation.





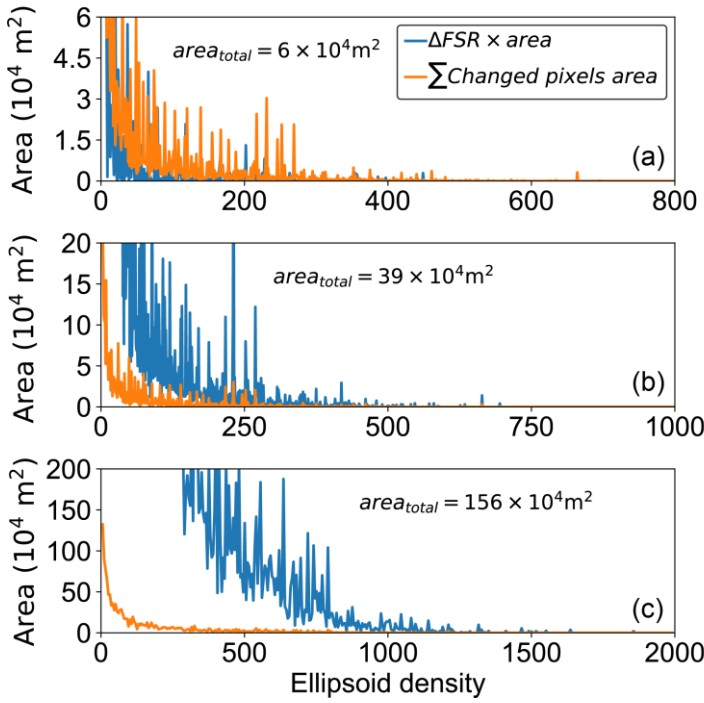

**Figure 7: Landslide density tests for tiles with (a) $20 \times 20$, (b) $50 \times 50$, and (c) $100 \times 100$ grid cells. The total areas for the three scenarios are also presented. Two targets are computed during an interval of $\triangle d_s = 1$.**

### 4.3 Characteristics of rainfall and flood events


Provided the essential parameters and datasets are appropriately prepared for iHydroSlide3D v1.0, we choose June 20, 00:00 to July 15, 00:00 as the simulation period, which is defined by two factors: (i) the period must include the main rainstorm triggering the flood and slope failures; and (ii) the period should be longer than the observation period to exclude the effect of initial conditions (Zhang et al., 2016;Wang et al., 2020). As illustrated in Fig. 8, the rainstorm started around July 4, 00:00 and

reached the peak rate (exceeded 25 mm/h) within 5 hours, and lasted for about a day across the region. The peak discharge was observed a few hours after the peak-rainfall moment, reaching a value close to 1000 m³/s. The comparison between the modeled and observed discharge shows a generally good agreement with Bias=37.9%, NSEC=0.77, and CC=0.93, respectively (Fig. 8). The slightly large bias implies there is likely some uncertainty in routing or flow concentration processes depicted by the hydrological module in iHydroSlide3D v1.0. Moreover, the model behaves sensitive to the rainfall data (before July 4,

02:00 and after July 7, 14:00). As a result, uncertainty in the rainfall data may contribute to the bias in the simulated stream flow. Nevertheless, the above results indicate that the iHydroSlide3D v1.0 is generally capable of simulating the flood events and runoff processes when the model is calibrated.





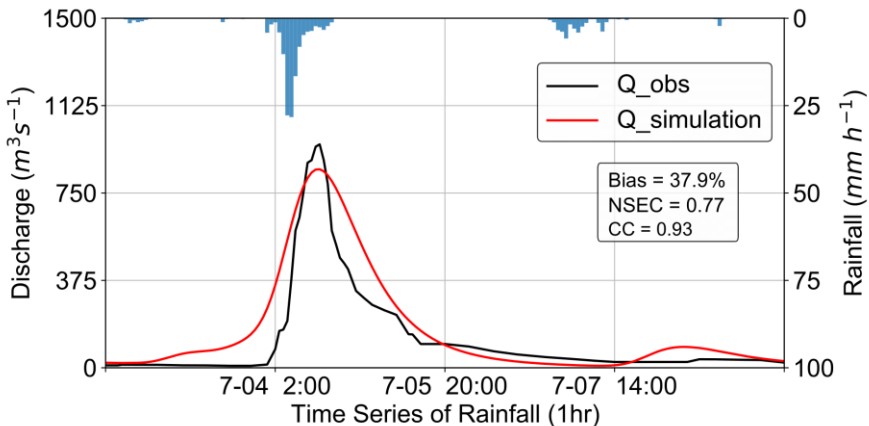

**Figure 8: Basin-average rainfall rates and modeled hydrographs against the observed streamflow.**

**4.4 Evolution of landslide risk responding to hydrological process**

**4.4.1 Soil moisture and factor of safety**

The evolutions of the soil moisture and landslide susceptibility are illustrated in Fig. 9. There is a small part of this region being predicted as unstable areas (Fig. 9a) in the beginning of the storm and can be explained by (i) the effect of antecedent rainfall or initial hydrological conditions, and (ii) some grid cells that have steep slopes and are extremely unstable (Arnone et

al., 2011;Aristizábal et al., 2016). These grid cells, generally located on very steep slopes, are more easily calculated as unstable areas in terms of $FS$ value according to Eq. (19), which may bring some overestimation in the iHydroSlide3D v1.0. However, we have attempted to avoid such weakness by using wetting front concept with regard to slope failure depth (Eq. (26)), which is subject to hydrologic infiltration process and remains very small at the early stage of the rainfall event. As a result, a very small portion is estimated (Fig. 9a). The soil moisture drastically increases when the rainstorm starts, particularly for the

computational elements (streaks in Fig. 9a and b) belonging to main routing channels of the drainage network. Based on the cell-to-cell flow routing rule, at the early stage of the storm, these cells have more chances to experience re-infiltration of excess surface runoff from upstream cells. As a consequence, they are more likely to reach a saturated condition. This phenomenon emphasizes the contribution of topography to the evolution of soil moisture at the early stage of a rainstorm, when the saturated hydraulic conductivity is relatively similar. In accordance with the soil moisture, more conditional unstable

grid cells are predicted compared to the spatial pattern before rainfall starts. Soil moisture and landslide risk still continue to increase 3 hours later and after the rainstorm reaches its peak; as a result, most of the study area is fully saturated and unstable cells are substantially increased (Fig. 9c). Different from the early stage, the excess portion of rainfall cannot effectively be absorbed by soil anymore but contributes to runoff instead, leading to the flood along the river channel (Fig. 8). No significant difference can be observed between Fig. 9c and d, as the water amount of the rainfall has exceeded the infiltration demand and

water capacity.



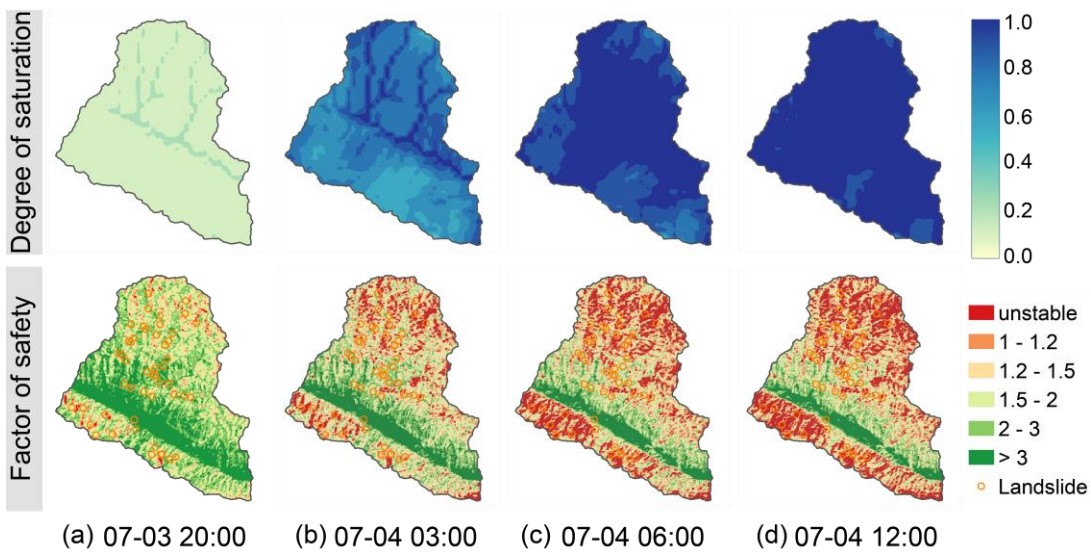

**Figure 9: Spatiotemporal evolutions of the soil wetness (*i.e.*, degree of saturation) and factor of safety. (a)~(d) are four moments that span the complete rainfall event.**

### 4.4.2 Probability of failure

The model estimates the probability of failure for each grid cell due to the random operation of potential landslide generation (Sect. 2.7), although soil properties and hydrological conditions are deterministic. The original unstable areas were further re-classified to a different degree of probability (Fig. 10). We specifically divided the risk zones in terms of the *PF* values referring to the available classification (Lizárraga and Buscarnera, 2020;Vandromme et al., 2020): low ($0 < PF < 5\%$), moderate ($5\% < PF < 30\%$), high ($30\% < PF < 60\%$), and very high ($PF > 60\%$). As shown in Fig. 10a, most of the

unconditional unstable areas fell in zones of low and moderate susceptibility whilst the others were estimated as the risks of high or very high. The former grid cells (*e.g.*, inset 2 in Fig. 10), affected by the cell with a steep slope, might be computed as unstable because iHydroSlide3D v1.0 assesses the slope stability using the 3D landslide model (Eq. (19)) and then outputs the minimum *FS* after random tests. In this work, relying on the *PF* classification, we can infer there are only a few steep grid cells (includes themselves) near the grid cells with small values of *PF*, at least they are attenuated by the flat terrain. On the

other side, for the grid cells with large *PF* values (*e.g.*, inset 1 in Fig. 10, zones of high or very high), the local topography is more likely to be continuous steep slopes that can be repeatedly calculated as unstable and thus cause larger *PF* values (Eq. 28). However, very few landslides are observed in the areas with steep slopes (Fig. 9 and 10). These areas may be covered by no or very thin colluvium or regolith; under this circumstance, soil depth tends to be negatively related to slope angle according to field survey or available soil thickness models (Ho et al., 2012;Lanni et al., 2012;Alvioli and Baum, 2016;Tran et al., 2018).

In this way, hazards like rockfall or avalanche are more expected instead of rainfall-induced landslides for these areas with extremely steep slope angles. Spatiotemporal evolution of the *PF* value shows that the probabilistic approach is capable of not only identifying the stable or unstable areas but also monitoring the unstable area in a more reliable and informative way.



Compared to the binary assessment (stable & unstable), this method can help to better understand the relationship of landslide risk with local topography and dynamic hydrological conditions.

iHydroSlide3D v1.0 depicted the evolutions of unstable area and all risk zones (in percent of the whole region) introduced above over the computational time (curves in Fig. 11). These two areas are controlled by the patterns of $FS$ and $PF$, respectively. Overall, the unstable area holds its leading position during the complete rainstorm. More specifically, $FS$ values respond more dramatically to the rainfall event than $PF$ values, which makes the unstable area increase more rapidly at the peak stage of the rainfall. This is not surprising because changing the value of $PF$ should obey stricter rules (Eqs. (27) and

(28)) and experience repeatedly random tests. Among the various classes of the probability, the percent area and sensitivity to rainfall decrease with increasing $PF$-class value (see in Fig. 11). At the early stage, the unconditional unstable area is computed less than 5%, followed by percent area according to the $PF$ values, particularly for $PF > 60\%$ (close to zero, precisely 0.12%). At the end of the rainfall (the soil is nearly fully saturated and the curves are steady), the percent area with $PF > 5\%$ is about 10% less than the total unstable area, followed by the other zones of risk. A slight increase is observed for $PF > 60\%$ (zone

of very high) and most of them are contributed by unconditional unstable, which is immune to hydrological process (Aristizábal et al., 2016). The rest of the curves lie between them. The spatiotemporal classification of the landslide probability, as well as the traditional binary state of slope stability, are meaningful for landslide risk delineation and monitoring the area with a specific failure probability of interest.

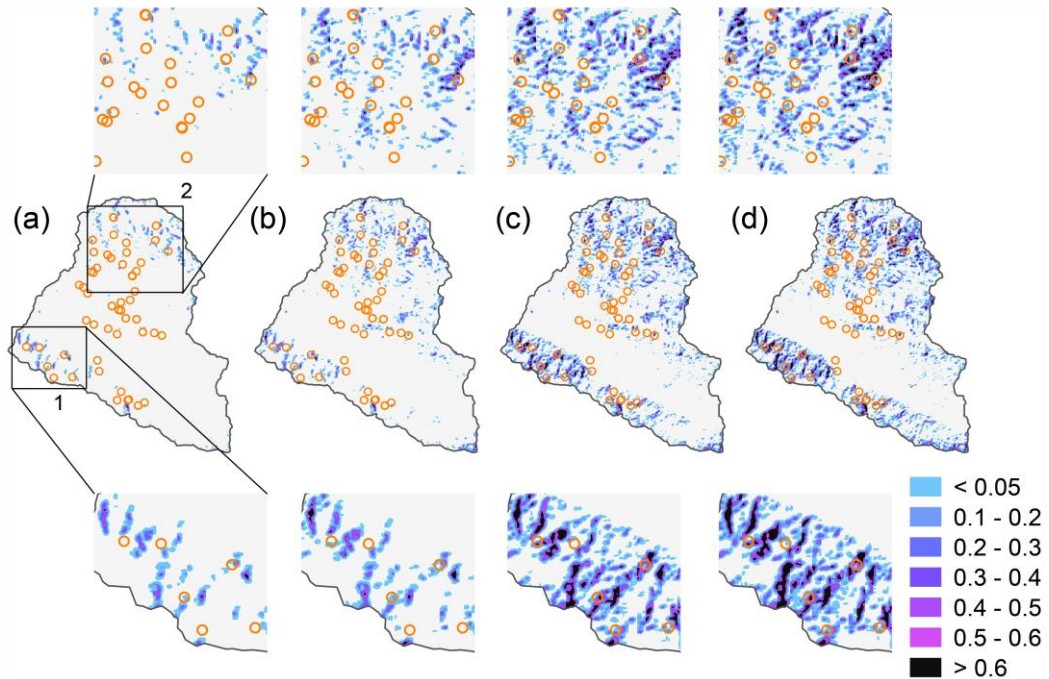

**Figure 10: Spatiotemporal evolutions of the landslide occurrence probability. (a)~(d) are four moments that span the complete rainfall event.**





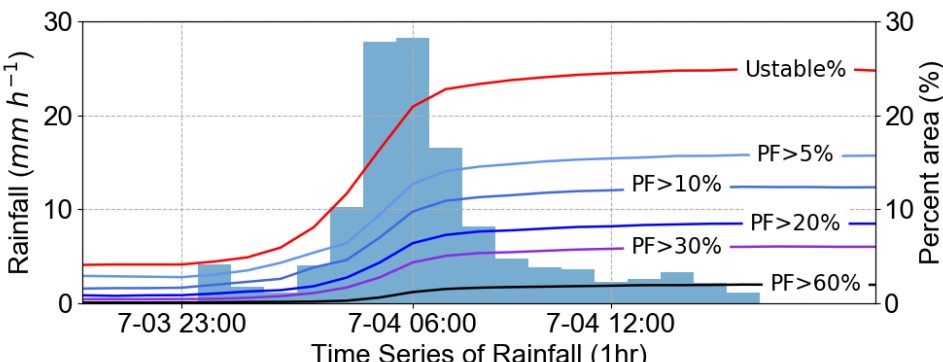

**Figure 11: Evolution of percent area computed as unstable or various failure probabilities as the rainfall continues.**

**4.5 Spatial performance of model**

We evaluated the spatial performance of the iHydroSlide3D v1.0 during the study period as presented in Table 5. We also compare our model with the previous coupled model CRESLIDE, of which the infinite slope stability is adopted (Fig. 12). Results show that 33 out of 54 landslides were successfully predicted, falling into the area with $FS < 1$ and $PF > 1$. For the zones of landslide risk, most of the failures (reaches 53.7%) are observed in low and moderate risk zones, whilst the remainder are in the zones with high and very high risks. The value of %$LRclass$ index is evaluated as 82.91% when using factor of

safety for prediction, and the same index reaches 94.05% when we add up the values for all four risk zones. To be less conservative, the %$LRclass$ index for $PF$ prediction can be 82.79%, which is close to the value by $FS$ prediction, if we only consider the landslide risk from low to high. This result can be explained by the number of landslides per unit area, *i.e.*, the binary approach would cover more extensive areas to contain the landslide locations. By adopting the probabilistic approach to identify classified risk zones, we can focus on the area of interest and make more targeted and efficient predictions.

520           The ROC analysis demonstrates that the iHydroSlide3D v1.0 generally has a higher hit rate and lower false positive rate relative to the CRESLIDE model that is coupled with the infinite landslide model. The Area Under the ROC Curve (AUC) values for them are 0.77 and 0.72, respectively, suggesting that iHydroSlide3D v1.0 outperforms CRESLIDE in this case study. As mentioned in Sect. 2.3.2, the most significant difference between the two models is the assumption of landslide geometry. The 3D model takes the neighbouring cells into account and thus provides a comprehensive $FS$ value (Eq. (19)), while the

infinite models abruptly solve the limit equilibrium equation on a solo raster cell and are strongly conditioned by the local topography (Mergili et al., 2014b). This explains why the infinite-type models have a tendency to provide more conservative results (*i.e.*, lower stability or worst situation) (Xie et al., 2006;Tran et al., 2018;Mergili et al., 2014b;Chakraborty and Goswami, 2016;He et al., 2021), indicated by higher false positive rates (*e.g*., 0.32 for CRESLIDE versus 0.20 for iHydroSlide3D when the threshold equals 1) in this study.


**Table 5: Comparison of _LRclass_ and %_LRclass_ obtained from _FS_ and _PF_ values. The unstable areas are further divided into several risk zones with regard to their _PF_ values.**

| FS class | Number of events ($a$) | Ratio to total events $(c = \frac{a}{b})$ | % predicted area $(d = \frac{cell_{class}}{cell_{total}})$ | _LRclass_ $(e = \frac{c}{d})$ | %_LRclass_ $(= e/f)$ |
|---|---|---|---|---|---|
| FS < 1 | 33 | 61.11 | 24.46 | 2.50 | 82.91 |
| FS > 1 | 21 | 38.89 | 75.54 | 0.51 | 17.09 |
| Total events | 54 ($b$) | 100 | 100 | 3.01 ($f$) | 100 |
| $PF = 0$ (Null) | 21 | 38.89 | 75.54 | 0.51 | 5.95 |
| $0 < PF < 5\%$ (Low) | 13 | 24.07 | 9.04 | 2.66 | 30.76 |
| $5\% < PF < 30\%$ (Moderate) | 16 | 29.63 | 9.6 | 3.09 | 35.66 |
| $30\% < PF < 60\%$ (High) | 3 | 5.56 | 3.92 | 1.42 | 16.37 |
| $PF > 60\%$ (Very high) | 1 | 1.85 | 1.9 | 0.97 | 11.26 |
| Total events | 54 | 100 | 100 | 8.66 | 100 |

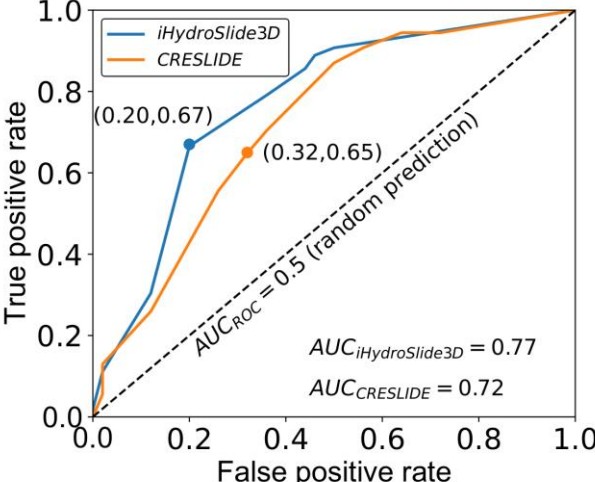


**Figure 12: ROC plot comparing slope-stability results from the CRESLIDE and iHydroSlide3D v1.0 models. The points on curves correspond to $FS = 1$ for both models. The AUC values are also shown in the plot.**

**4.6 Landslide hazard analysis**

The iHydroSlide3D v1.0 is capable of computing the extent (_i.e._, the area $A_L$ and volume $V_L$) of potential landslides, which is

essential for landslide hazard assessment. Compared to the visual techniques (_e.g._, aerial photograph interpretation and high-resolution imagery) or _in-situ_ investigation, the model estimates the $A_L$ and $V_L$ in a physics-based manner and strongly depends on the restrictions of random ellipsoids. In this way, $A_L$ is simply determined by the number of encompassed raster cells, while





$V_L$ is computed by the soil columns and the failure depth associated with hydrological infiltration (Eq. 26). Therefore, there exist common phenomena that the values of $V_L$ are more variable than that of the $A_L$, *i.e.*, one unique $A_L$ may correspond to

multiple $V_L$. Further, the adjacent cells may share the same value of $A_L$ and $V_L$ because they are possible to fall into the same potential landslide. In this work, we recorded and presented the max value of the $A_L$ and $V_L$ as the worst scenario across the unstable area (see in Fig. 13) after the sufficient random tests. Results show that most of the areas range from $4 \times 10^4 \text{m}^2$ to $5 \times 10^4 \text{m}^2$, while the volumes are more variable with a maximum value of around 1.1 million $\text{m}^3$. The relatively large value of $V_L$ may be resulted from (i) a relatively large $A_L$ that contains more soil columns or (ii) deep-seated landslides involved. It

is worth noting that the areas with extremely large values of $V_L$ (Fig. 13b) are roughly overlapped by the areas with relatively large $PF$ (Fig. 10d). This can be explained by that, in our pursuit of the minimum of the $FS$, a relatively thick failure depth was adopted in these areas, which caused an overprediction for landslide areas (Ho et al., 2012). Although the maximum magnitudes ($A_L$ and $V_L$) of landslide hazards provide more conservative assessments, we expect that they are acceptable in slope engineering assessment (Tran et al., 2018).

Due to lack of historical documents for real $A_L$ and $V_L$ in this field, we evaluate the landslide hazard results by fitting the relationships of the $A_L$ and $V_L$ and comparing them with the existing relationships reported in previous literature. As the nature of these two geometrical properties introduced above, we did not collect all the values for each pixel. Here we prepared the fitted source into six data sets according to the combinations of $A_L$ and $V_L$ (source data in Table 6). All possible $V_L$ values referred to the cases with $PF_{min}$ and $PF_{max}$, and four risk zones. We further fitted these six sets by power law and counted the

R-square number (see in Table 6). Moreover, as a comparison, we collected four available relationships from previous literature computed using field measurements in their study (Table 6, ID 7 to 10). We then plotted them by substituting the $A_L$ values in this work (See in Fig. 14). Obviously, relatively less data is plotted in Fig. 14a and b, which, as we have pointed above, shows all possible areas for potential landslides without duplicate value. The values of $V_L$ estimated with $PF_{max}$ (Fig. 14b) are relatively larger than that with $PF_{min}$ (Fig. 14a) because the deeper slip depth tends to obtain a smaller $FS$, which in turn

inevitably results in a larger volume of a failure. The fitted curves are close to the available equations in terms of trend, among which the Abele (1974) model overestimated the $V_L$ in cases with ID 1 and 2. The efficiency of the fitted equations is generally good in terms of $R^2$, reaching 0.992. However, such a power model has low efficiency for cases of ID 3 to 6 with low $R^2$ and abnormally wide confidence intervals. Although these cases adopt the unique combinations of $A_L$ and $V_L$, it is still very likely to accept the samples with identical $A_L$ and consequently get more dots in $A_L \sim V_L$ graph (Fig. 14c, d, e, and f), which further

pose hinders to fit them as functions (*i.e.*, a binary relation between two sets that associates every element of the first set to exactly one element of the second set). In other words, they are regarded as sampling error when the power model is considered. We acknowledge that, in this work, we can only provide relatively ideal geometrical information (with regular and limited characteristics) in a mathematical manner, which is determined by cell size and random procedure. Even so, we appropriately consider the power models in the cases of ID 1 and 2 where unique values of $A_L$ are applied. We further note that such

relationships are not only limited to the maximum and minimum $PF$ value but also any value of interest on the users' side. For





those applications limited by field measurements, the method proposed here is expected to roughly assess the magnitude of landslide hazards.

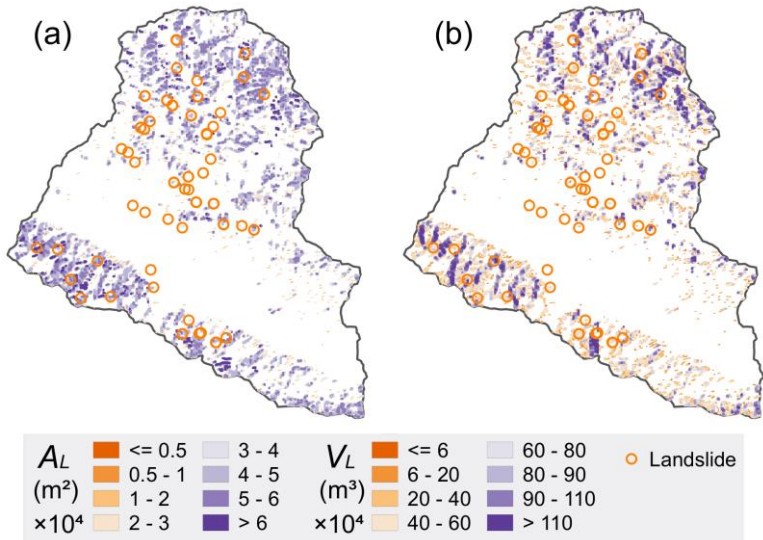

**Figure 13: Spatial patterns of the max values of (a) $A_L$ and (b) $V_L$ for model-predicted landslides.**

**Table 6: Relationships linking maximum landslide area $A_L$ to landslide volume $V_L$.**

| ID | Equation | $A_{Lmin}(m^2)$ | $A_{Lmax}(m^2)$ | Source data used to fit | $R^2$ |
|---|---|---|---|---|---|
| 1 | $V_L = 285.5 \times A_L^{0.687}$ | $5.47 \times 10^3$ | $9 \times 10^4$ | Unique $A_L$ and $V_L$ with $PF_{min}$ | 0.992 |
| 2 | $V_L = 146.4 \times A_L^{0.766}$ | $5.47 \times 10^3$ | $9 \times 10^4$ | Unique $A_L$ and $V_L$ with $PF_{max}$ | 0.992 |
| 3 | $V_L = 26.727 \times A_L^{1.061}$ | $5.47 \times 10^3$ | $9 \times 10^4$ | Unique combination of $A_L$ and $V_L$ in the zone of low | 0.599 |
| 4 | $V_L = 80.29 \times A_L^{0.842}$ | $7.19 \times 10^3$ | $8.83 \times 10^4$ | Unique combination of $A_L$ and $V_L$ in the zone of moderate | 0.184 |
| 5 | $V_L = 513.4 \times A_L^{0.684}$ | $6.25 \times 10^3$ | $8.83 \times 10^4$ | Unique combination of $A_L$ and $V_L$ in the zone of high | 0.13 |
| 6 | $V_L = 154.1 \times A_L^{0.806}$ | $6.25 \times 10^3$ | $8.58 \times 10^4$ | Unique combination of $A_L$ and $V_L$ in the zone of very high | 0.221 |
| 7 | $V_L = 0.074 \times A_L^{1.450}$ | $2 \times 10^0$ | $1 \times 10^9$ | Guzzetti et al. (2009) | |
| 8 | $V_L = 0.39 \times A_L^{1.31}$ | $1 \times 10^1$ | $3 \times 10^3$ | Imaizumi and Sidle (2007) | |
| 9 | $V_L = 0.242 \times A_L^{1.307}$ | $2 \times 10^5$ | $6 \times 10^7$ | Abele (1974) | |
| 10 | $V_L = 12.273 \times A_L^{1.047}$ | $3 \times 10^5$ | $3.9 \times 10^{10}$ | Haflidason et al. (2005) | |

Column 1 lists the equation number. Column 2 shows the fitted equations in this work (ID 1 to 6) and available equations (ID 7 to 10) selected from previous literature. Columns 2 and 3 list the ranges of $A_L$ applied for equations; the data for ID 1 to 6 is





from this work; data for ID 7 to 10 is from literature. Column 4 gives the data source. Column 5 lists the commonly statistical measure R-squared ($R^2$).

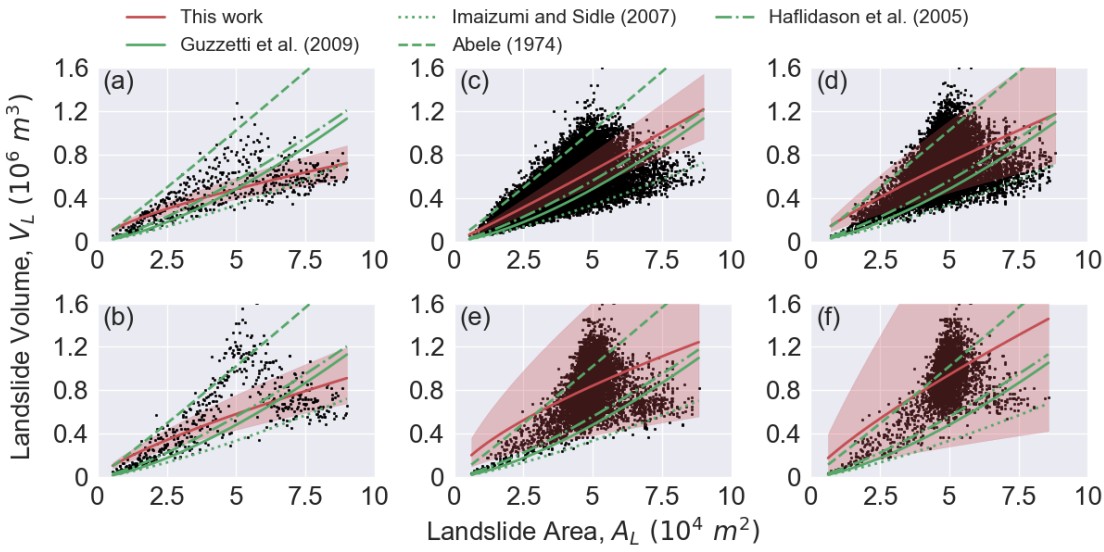

**Figure 14: Six sets of source data (ID 1 to 6 in Table 6) are plotted and fitted in this work. All available equations (ID 7 to 10 in Table 6) are plotted by substituting the $A_L$ values in this work. Red zone shows 95% confidence intervals.**

## 5 Discussion and conclusions

We have modified the 3D landslide model to make it applicable for more general situations (*i. e.*, all possible soil moisture state). To this end, we incorporated the distributed hydrological model CREST to undertake the computational task of hydrological components, forming a new coupled hydrological-geotechnical model called iHydroSlide3D v1.0. The model is capable of assessing the spatiotemporal landslide susceptibility ($FS$ and $PF$), performing hazard analysis (geometric properties of landslides, $A_L$ and $V_L$), and predicting flash floods driven by rainfall processes. Considering differential needs for computational resolutions by the hydrological and landslide modules, we embedded the soil downscaling method to seamlessly execute the code within such a sophisticated framework containing two resolutions datasets. For the purpose that the code is practicably performed in the case of large scale and, meanwhile, the computational time is at an acceptable level, we program the code in a parallel manner and run it on a multi-core machine. We then tested and evaluated the model in a region suffering recorded rainstorm and slope failures.

Prior information on parameters is necessary for this model and need to be handled with the utmost care. As a matter of fact, most of the parameters are determined by available datasets and field records, while few of them are calibrated manually based on computational experimental tests. In particular, we want to point out that landslide density could significantly affect the output results and, even worse, a small value may yield meaningless results and unwanted consequences. Thus, the landslide density is necessary to be regularly tested when the code is applied for a new region. However, we would preliminarily recommend $d_s = 500$ for a rough assessment as it has been tested in detail in this study and a study by Mergili



et al. (2014a). We conclude that the converged density value tends to be irrelevant to the tile area once the constraints of the landslide's shape are determined. We also argue that the soil downscaling method is necessary when we run the hydrological and landslide modules at different resolutions, because the uneven soil moisture patterns exactly impact the slope stability assessment. In particular, the 3D stability model should sufficiently consider the spatial distribution of soil moisture within an objective slip surface. This is a typical difference when we adopt the downscaling method comparing to the infinite stability

model (Wang et al., 2020).

       In this work, we have prepared the observed river streamflow from the gauge and the point-like landslide locations. Although we have gotten a generally good agreement with the observations in terms of discharge and similar efforts have been done in previous studies (He et al., 2016;Zhang et al., 2016;Wang et al., 2020), the results cannot directly prove that the soil moisture is accurately estimated, which is truly associated with slope stability, per se. Other soil moisture data through site

measurement (Lepore et al., 2013) or satellite (Zhuo et al., 2019a;Zhuo et al., 2019b) can be used to further validate the model performance. However, field measurements are usually not available and even many boreholes can only cover some of the many grid cells in a large-scale region (Marin et al., 2021), making the representativeness of ground observations questionable. The observation from the satellite is useful for soil moisture in shallow depth, hindering the application for landslide predictions at a deep depth (Zhuo et al., 2019a). Therefore, we consider the soil moisture as an intermediate hydrological component, of

which the spatial pattern is simulated at each time step.

       The model advantageously provides a spatiotemporal perspective for the evolution of hydrological processes, as well as the landslide assessments and hazards. Together with the random operation, the model can simultaneously assign the unstable grid cells with factor of safety and failure probability. We expect such a combination of landslide assessment analysis is effective and more targeted. Moreover, temporal monitoring of the process evolution is useful for dynamic management of

unstable areas subject to rainfall events. The overall performance of the model is generally satisfactory based on the statistical metrics of both hydrological (Bias, NSEC, CC) and landslide aspects ($\%LR_{class}, ROC - AUC$). We further recommend that the $\%LR_{class}$ index can be appropriately used to evaluate the landslides within various zones of risk determined by $PF$ ranges. Note that we did not distinguish the unconditional stable and unstable grid cells beforehand, although they can inherently occur in the landslide models built upon the limit equilibrium principle (Aristizábal et al., 2016). However, iHydroSlide3D v1.0

defined the failure depth by adopting the wetting front concept that is subject to the infiltration process. The model, therefore, can better target the rainfall event and reasonably handles the hydrologic initial conditions. In addition, the results also indicate that the 3D landslide model can ameliorate the overprediction problem, known to be present in the infinite landslide models.

       In the present work, we produced the geometric properties of potential landslides; however, the verification of results is still limited by the available measured data (*e.g.* landslide scars used in Arnone et al. (2011)). Instead, we evaluated them

with the fitted power-law equations, which, together with the available relationships in previous studies, are used as statistical tools for analysis of regional landslide magnitude. As a matter of fact, we haven't unveiled the fundamental geotechnical mechanics of landslide in terms of 3D geometry of the sliding surface, which need be solved through field investigation. The



work we have done here is similar to that of Marchesini et al. (2009) based on the limit equilibrium method and iteration. What makes progress is that we perform the model in a large region and obtained more detailed results.

Another limitation is the geotechnical parameters extracted from the available datasets. Determining their values in this way cannot consider geotechnical uncertainty due to inherent temporal and spatial variability of terrain materials (Hicks and Spencer, 2010;Griffiths et al., 2011;Mergili et al., 2014a). One way to overcome the problem is adopting the Monte Carlo approach, of which the examples can be found in literature (Raia et al., 2014;Mergili et al., 2014a;Vandromme et al., 2020). Such embedded probabilistic method, no doubt, will considerably bring additional computational burden. In addition, we

associate the failure depth with the infiltration process in this work, neglecting the spatial distribution of soil thickness in a terrain, which shall be a subject of future studies by supplying different soil-thickness assumptions.

In summary, a new hydrological-geotechnical model, iHydroSlide3D v1.0, coupling a distributed hydrological model (CREST) and a three-dimensional slope stability model (3D landslide model), was described and tested in this study. The model is capable of simulating the spatiotemporal evolutions of hydrological components and landslide susceptibility and

hazard. In order to coordinate the different resolution of datasets required for hydrological and landslide modules, the soil downscaling module is embedded to ensure that the code can be seamlessly executed. For efficiency, we program the code within a parallel framework and, together with the auxiliary efforts, make it possible to run in a large region. The model comprehensively presented the consequences of rainfall-triggered landslides at the watershed scale. With the evaluations from both hydrological and landslide aspects, we highlight the performance of iHydroSlide3D v1.0 on back-analysis and the

potential for predicting cascading flood–landslide disasters. The produced zones of risk and landslide geometric properties are valuable for disaster prevention and risk management. The modelling system presented in this work is also designed as a framework and has the potential to adopt other hydrological or land surface model (LSM) schemes and landslide models as alternatives. Moreover, iHydroSlide3D v1.0 can be further improved by optimizing geotechnical parameters and adopting other soil-thickness assumptions.


*Code and data availability*. The source code to iHydroSlide3D v1.0 is available on GitHub at https://github.com/GuodingChen/iHydroSlide3D_v1.0/tree/v1.0  and on Zenodo at https://zenodo.org/record/4577536 with a DOI of http://doi.org/10.5281/zenodo.4577536. The data of results displayed in this paper are provided, along with the plot code, on GitHub at https://github.com/GuodingChen/Data-Plot_code/tree/Data&plot_code and on Zenodo at https://zenodo.org/record/4559938 with a DOI

of http://doi.org/10.5281/zenodo.4559938.

*Author contributions*. KZ and GdC designed this study; GdC, KZ and SW developed the model code and performed the simulations; LjC provided the original datastes; GdC and KZ prepared the manuscript with contributions from all co-authors; KZ acquired fundings for this study.


*Competing interests*. The authors declare that they have no conflict of interest.



*Acknowledgements.* This study was supported by the National Natural Science Foundation of China (51879067), the National Key Research and Development Program of China (2018YFC1508101), the Natural Science Foundation of Jiangsu Province
(BK20180022), Six Talent Peaks Project in Jiangsu Province (NY-004), and the Fundamental Research Funds for the Central Universities of China (B200204038).

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
