# Peer review of "iHydroSlide3D v1.0: an advanced hydrological-geotechnical model for hydrological simulation and three-dimensional landslide prediction"

_Geoscientific Model Development, 2021_

## Author Response (AR1)

**Reply to all comments**

We highly appreciated the comments from the referee and thank the editor for managing this manuscript. Our responses and changes to all the comments are listed as follows:

**Comment:** The scientific and technical contribution of the paper and model is clear, but it would be good for the authors to mention the real-world applications of combined hydrological-geotechnical models for land planning and disaster risk management. Although briefly mentioned at the end (P29 L655-656), discussing the importance of this type of model to policy-makers and decision-makers would further underscore its utility.

**Response:** Thank you for the comment. Literatures have shown the contributions of hydrological-geotechnical models to real-world applications, such as improvements in disaster preparedness and hazard management in North Carolina, US (Zhang et al., 2016) and long-term vulnerability estimates in Shaanxi Province, China (Wang et al., 2020), to name a few.

**Changes:** We have mentioned it in the introduction (P2 L43). We also added a further discussion in the following section (P31 L656).

**Comment:** Section 2.8: this section needs some revision to present what inputs are needed in a more organized manner. The first sentence gives a high-level overview of what inputs datasets are needed, but the section doesn't give a good idea of what other parameters are needed in association with these datasets. P13 L337-338 then says that hydrological parameters are needed, but with only one example. Later in Section 3 (P14 L370), it says that information about the impervious surface area was calculated from the land cover map. This section would be more useful if it put the required underlying data and parameters up front. The following table is just a suggestion as a starting point.

**Response:** We agree with your suggestion. To improve it, the model inputs can be summarized into four types: meteorological forcing data, land surface feature data, simulation parameters, and calibration/verification data. The detailed description,

value/resolution, and source can be found in Sect. 3.

**Changes:** We have corrected it and added a new table (see Table 1, P13) to give an overview of the input datasets needed in iHydroSlide3D v1.0.

**Comment:** Like the introduction section, it will be good to have more discussion about how the model framework can contribute to land management and disaster risk management.

P29 L655-656 says: "The produced zones of risk and landslide geometric properties are valuable for disaster prevention and risk management."

Can you give some examples of how you see the model framework and code being used in these situations? Would they be used for climate change scenarios? Would they be used for real-time modelling? Would the risk zones be used as guidelines for no-build areas?

**Response:** We thank the reviewer for the useful comment. With professional analysis, comprehensive assessments (in both flood and landslide) may contribute to land management and disaster risk management. Landslide susceptibility and hazard zoning are able to manage landslide hazards in urban/rural areas by excluding development in higher-hazard areas and requiring hydro-geotechnical assessment in the planning stage (Fell et al., 2008). The conception has been introduced in countries such as France (Fell et al., 2008) and Switzerland (Leroi et al., 2005). A recent study corroborated existing hypotheses that urbanization increases landslide hazards (Johnston et al., 2021). Our model could be used as a tool to study the importance of considering interactions with urbanization when predicting landslide hazards under different climate change scenarios. The current modular framework and flexibility of the modeling setup also make it feasible to link with other numerical weather prediction models and real-time forcings. We would like to stress that these complicated applications generally require extraordinary computing resources to support.

**Changes:** We have added a new paragraph to discuss this point (see P31 L656).

**Comment:** P3 L90: "physically-based"?

**Response and changes:** We have corrected it and several other typos in the text.

**Comment:** P4 L100-101: Add access dates to URLs

**Response and changes:** The URL links have been added to them.

**Comment:** P10 L276: Why is the minimum value of FS and PF assigned to the cell?

**Response:** For regional modeling, our 3D landslide model randomly generates a large number of ellipsoidal potential landslides. As a result, each grid cell will participate in the stability analysis of various landslides multiple times, contradicting the value of FS (see in Fig. 4). The model did not provide the minimum PF but defined it by counting FS values (see Fig. 4 and Eq. (28)). We acknowledge that the model outputs the minimum value of FS (called $FS_{min}$ in this work and corresponding to the worst-case situation). However, we computed the PF value to support the analysis, which further divides unstable regions into different probability levels. In addition, our analyses for landslide risk zones (P23 L504) are also based on the various values of the PF.

**Changes:** We further clarified this point to avoid confusion (P11 L208 and P12 L305).

**Comment:** P11 L280: What is meant by "sufficiently large number of ellipsoids" and how is that determined? P16 Table 3 talks about the parameter set based on landslide inventory, but how is that calculated?

**Response:** In this work, a dimensional "landslide density" is defined to reflect the number of ellipsoids. Eq. (31) clarified the way to calculate the total number of tested landslides (i.e., the random sampling times). The landslide density thus serves as a calibrated parameter in iHydroSlide3D. We carried out several convergence tests to find a reasonable value for landslide density (Fig. 7). The so-called "sufficiently large number" means that a steady (or convergent) result has been reached. Specifically, the effect on the outcome is stronger when the density value is relatively low, and it becomes weaker as the value steadily rises. However, increased density will significantly increase computational overhead. As a result, we chose the value of 500 in this work.

The parameters set based on landslide inventory in Table 3 are the maximum and minimum landslide dimensions (length and width). Our inventory did not record the dimensional information for all landslides, but a few of them. We extract the maximum and minimum length and width from this limited dataset. As a result, the maximum/minimum values served as the constraints for landslide modeling. Considering the random interval equals the spatial resolution (12.5 m in this work), the constraint boundaries were rounded to the nearest integer for simplification.

**Changes:** We added the explanation at P11 L284 and P15 L381.

**Comment:** P11 Figure 4: The white letters and numbers are hard to see, would recommend a black outline to make them stand out.

**Response and changes:** Thanks for the reminder. We have modified it.

**Comment:** P14 L375: Please give an idea of how much time was needed, either for each module (hydrological and geotechnical) or total runtime to get the user an idea of computational efficiency.

**Response:** It's not easy to separately record the runtime for each module since the current version is executed in a coupled manner. The computing time for simulation is 55432 s (328 s for each time step). We should note that the runtime is highly dependent on computing resources/hardware (e.g., the CPU's performance, its core number, and its random-access memory (RAM)). We cannot indefinitely increase the number of computing cores, because the resulting memory requirements are often extremely expensive. Though parallel programming is adopted, this level of computing burden is still unaffordable for ordinary personal computers (PCs, normally with 8 cores and 16GB of RAM). We therefore consider high performance computing (HPC) an appropriate way to perform iHydroSlide3D v1.0. The users can still however execute it on PC if the runtime is out of the pursuit. Moreover, the runtime also fluctuates significantly with "landslide density" changes. The computational efficiency is always a trade-off between accuracy and speed.

**Changes**: We have added the total runtime at P19 L439; we also further discussed the

runtime at P29 L612, where the efficiency comparison with TRIGRS v2.1 and r.slope.stability was also made.

**Comment:** P16 Table 3: Although defaults are recommended later in the text, it would be good to give some ranges/ballpark figures for Ncores, Landslide Density, TotalTile.

**Response:** Thanks for the comments. For personal computers, we recommend Ncores ≤ 5, considering the limited RAM. If the error says "out of memory", decrease the value until the model can run. For users who have access to high-performance computing (HPC), this value can be flexibly customized. The default value of "Landslide Density" is 100, and we recommend a range of 50 to infinity. We note that the total runtime is sensitive to landslide density, which should be set with utmost care. In the present work, the landslide density equals 500 under the convergence tests (see Sect. 4.2). The default value of "TotalTile" is set to 40 and ranges from 20 to infinity. It's a good choice for users to set it divisible by "Ncores" to make better use of computing resources.

**Changes:** All of the above information is newly available on GitHub at **manual**.

**Comment:** P21 Figure 9: It is difficult to see the orange landslide circles, would recommend maybe black outline hollow circles instead of orange outline.

**Response and changes:** Thanks for the recommendation. We have changed the color.

**Comment:** P22 Figure 10: Are these the same four moments as in Figure 9? I would recommend putting the timestamps here too.

**Response and changes:** Yes, they are. We have added the time moments for them.

**Comment:** P26 Figure 13: It is hard to see the orange circles again, and they may blend in with the orange colours in the legend. Would again recommend black outline hollow circles.

**Response and changes:** Thanks for the recommendation. We have changed the color.

**Comment:** P29 L596: What was the computational time required and what is

considered "acceptable"?

**Response:** We have added content to give a general impression of computational efficiency by comparing with two parallel software packages, TRIGRS v2.1 (Alvioli and Baum, 2016) and r.slope.stability (Mergili et al., 2014). The runtime for the single time step is 328 s for the present code, while it is 110 s and 1900 s for TRIGRS v2.1 and r.slope.stability, respectively, in their descriptive literatures. Such a comparison is unfair because the runtime was not obtained under the same testing prerequisites. Moreover, differences in model structure prevent them from being treated equally. TRIGRS v2.1 uses a simple infinite-slope description and r.slope.stability does not include the hydrological simulation. We also did not include Monte Carlo simulation, which is used by r.slope.stability to consider the natural variability of the geotechnical characteristics of the soils. Instead, we determined them based on available datasets. We would also like to emphasize that the programming language has an impact on model performance. However, our fully vectorized programming style guarantees computational efficiency. The computation time is therefore considered acceptable for flood–landslide cascading disasters triggered by rainfall or thunderstorm events (usually with a short simulation duration). We mentioned the runtime of these models here only for the general impression, upon which users may estimate the computational cost based on their hardware and simulation scale.

**Changes**: We have added these explanations at P29 L614.

**Comment:** I tested the code available on Zenodo. The included manual is great to show a step-by-step of what needs to be done. It was relatively easy to get it up and running, but I ran into some errors in MATLAB because I had not installed some toolboxes (e.g. Mapping Toolbox, Parallel Computing Toolbox, Curve Fitting Toolbox). It would be good if the manual included the list of Toolbox dependencies in case the user has a limited installation of MATLAB.

Once the Toolboxes were installed, I was able to run the code without errors. I wasn't sure how to interpret the datasets in the Results section, and it would be good if the manual could give a brief rundown on what the results represent. The accompanying

text files (Outlet_Results, Outpix_03501000_Results, Outpix_03501000_Results_Statistics) were also empty.

**Response:** We appreciate this helpful feedback. The code does rely on several toolboxes in MATLAB. In addition to the most basic installation content, the software requires additional toolboxes: Curve Fitting Toolbox, Global Optimization Toolbox, Optimization Toolbox, Partial Differential Equation Toolbox, Statistics and Machine Learning Toolbox, Symbolic Math Toolbox, MATLAB in the Cloud, MATLAB Parallel Server, and Parallel Computing Toolbox. For users who have most of the toolbox installed by default, they can run the software easily. For users who have a limited installation of MATLAB, they are advised to check the availability of toolboxes in advance. Users can also directly run the code without concerns and install the toolboxes needed according to the error prompt.

The empty text file is a small bug with csv file writing, and we have fixed it in our updated version of **iHydroSlide3D_v1.0**. The model provides results for the outlet location, which are written to csv format series file. The model will automatically calculate the output pixel and output the csv file named **"Outlet_Results"**. Users can also select any of the specified output pixels as long as they are within the specified watershed extent. For this purpose, users can define it in **"ControlFile"**. In our example, we chose the same pixel location as the basin outlet (named "Yuehe"). As a consequence, the result file for the output pixel location will be named **"Outpix_Yuehe_Results"**. If the observed streamflow is provided in the OBS fold, three statistical metrics Nash–Sutcliffe coefficient of efficiency (NSCE), Pearson correlation coefficient (CC), and relative bias will be computed and stored in **"Outpix_Yuehe_Results_Statistics"**. Please note that abnormalities may exist in statistics if streamflow data are missing or all zeros, for which we would like to suggest a simulation including a complete flooding event.

**Changes**: We have updated the software **manual** to include the list of Toolbox dependencies in the MATLAB environment. We have added the explanation for all possible outputs in our updated **manual**. A small bug of csv file writing was fixed, and

the code has been synchronized with the link in the "Code and data availability" section.

**Reference**

Alvioli, M., and Baum, R. L.: Parallelization of the TRIGRS model for rainfall-induced landslides using the message passing interface, Environmental Modelling & Software, 81, 122–135, 10.1016/j.envsoft.2016.04.002, 2016.

Fell, R., Corominas, J., Bonnard, C., Cascini, L., Leroi, E., and Savage, W. Z.: Guidelines for landslide susceptibility, hazard and risk zoning for land-use planning, Engineering geology, 102, 99-111, 2008.

Johnston, E. C., Davenport, F. V., Wang, L., Caers, J. K., Muthukrishnan, S., Burke, M., and Diffenbaugh, N. S.: Quantifying the effect of precipitation on landslide hazard in urbanized and non-urbanized areas, Geophysical Research Letters, 10.1029/2021GL094038, 2021.

Leroi, E., Bonnard, C., Fell, R., and McInnes, R.: Risk assessment and management, Landslide risk management, 159198, 2005.

Mergili, M., Marchesini, I., Rossi, M., Guzzetti, F., and Fellin, W.: Spatially distributed three-dimensional slope stability modelling in a raster GIS, Geomorphology, 206, 178–195, 10.1016/j.geomorph.2013.10.008, 2014.

Wang, S., Zhang, K., van Beek, L. P. H., Tian, X., and Bogaard, T. A.: Physically-based landslide prediction over a large region: Scaling low-resolution hydrological model results for high-resolution slope stability assessment, Environmental Modelling & Software, 124, 104607, 10.1016/j.envsoft.2019.104607, 2020.

Zhang, K., Xue, X., Hong, Y., Gourley, J. J., Lu, N., Wan, Z., Hong, Z., and Wooten, R.: iCRESTRIGRS: A coupled modeling system for cascading Flood–Landslide disaster forecasting, Hydrology and Earth System Sciences, 20, 5035–5048, 10.5194/hess-20-5035-2016, 2016.

---

## Referee Report (RR1)

**Review of updated manuscript for: iHydroSlide3D v1.0: an advanced hydrological-geotechnical model for hydrological simulation and three-dimensional landslide prediction**

**Overview**

The revised manuscript is generally more organised, clear, and outlines the real-world applications of combined hydrological-geotechnical models for land planning and disaster risk management. shows the utility of the model, and its contributions to this field of modelling. The paper gives good insight as to how the probabilistic approach of landslide risk in iHydroSlide3D v1.0 is advantageous over conventional binary assessments of landslide risk. This revised manuscript more clearly outlines of the assumptions, equations, inputs, suggested parameters, novelty, and limitations of the model. I only have minor comments around some phrases and terms that can be reworded or defined to improve clarity.

**Specific comments**

P1 L16: "The model features the feasibility of applying flexibly different simulating resolutions for hydrological and slope stability submodules by embedding a soil moisture downscaling method."

- Suggest rewording this sentence for clarity. Maybe:
  - "Through embedding a soil moisture downscaling method, this model is able to model hydrological and slope stability submodules even at different resolutions"

P2 L35: Change "Modelling of landslide…" to "Modelling landslide…"

P2 L43: Change "Literatures have shown…" to "Literature has shown…"

P2 L58 to L60: "As a matter of fact, to the best of our knowledge, there are still very few fully coupled hydrological-geotechnical models that are capable of performing in a large scale and producing 3D information of landslide disasters."

- Suggest making this more concise:
  - "To date, there are still very few fully coupled hydrological-geotechnical models capable of performing at large scales and producing 3D information of landslide disasters."
- There are other sentences in the paper that could also use further editing to make them more concise. Phrases like "we want to point out" or "as a matter of fact" can be removed.

P2 L61 to L62: "Another problem that will be involved is the selection of computational spatial resolution."

- Suggest making this more concise: "Another problem is the selection of…"

P3 L66: "squared meters" to "square meters" or "meters squared"

P4 L102 to L103: If URLs are placed in the text, please add date of access.

P9 L237 to L239: The paragraph mentions that TWI is related to soil moisture through the wetness coefficient, and the reader is directed to another paper. Please briefly explain this relationship to the reader in this section, just enough for them to understand the basics.

P19 L446: Change "… the model behaves sensitive…" to "… the model is sensitive…"

P28 L605: Change "… containing two resolutions datasets…" to "… containing two datasets with different resolutions…"

P28 L605 to L606: "For the purpose that the model is practicably performed in the case of large scale, we parallelized the program for efficiency."

- Suggest rewording this sentence for clarity, particularly the first part.

P28 L609: Change "literatures" to "literature."

P28 L614: Remove "as a matter of fact"

P28 L616: Remove "we want to point out"

P30 L670: There seems to be a strange tab/space here in the manuscript, maybe "In summary…" was supposed to be part of a new paragraph?

---

## Author Response (AR2)

**Reply to comments**

We highly appreciate the comments from the referee and thank the editor for managing our revised manuscript. Our responses and changes to all the comments are listed as follows (the page and line of the modified location correspond to the tracked version):

**Comment:** P1 L16: "The model features the feasibility of applying flexibly different simulating resolutions for hydrological and slope stability submodules by embedding a soil moisture downscaling method." Suggest rewording this sentence for clarity.

**Response and changes:** Thanks for the comment. As suggested, we have changed this sentence to "Through embedding a soil moisture downscaling method, this model is able to model hydrological and slope stability submodules even at different resolutions" for better clarity. (P1 L18)

**Comment:** P2 L35: Change "Modelling of landslide…" to "Modelling landslide…"

**Response and change:** The correction has been done. (P2 L36)

**Comment:** P2 L43: Change "Literatures have shown…" to "Literature has shown…"

**Response and change:** The correction has been done. (P2 L43)

**Comment:** P2 L58 to L60: "As a matter of fact, to the best of our knowledge, there are still very few fully coupled hydrological-geotechnical models that are capable of performing in a large scale and producing 3D information of landslide disasters." Suggest making this more concise.

**Response and changes:** Thank you for the comment. We have simplified this sentence to "To date, there are still very few fully coupled hydrological-geotechnical models capable of performing at large scales and producing 3D information of landslide disasters.". (P2 L61)

**Comment:** There are other sentences in the paper that could also use further editing to

make them more concise. Phrases like "we want to point out" or "as a matter of fact" can be removed.

**Response and change:** We have removed redundant phrases to make the revised manuscript more concise (see P6 L170, P26 L587, P26 L589, P27 591, P28 L614, P30 L663, and P30 L667).

**Comment:** P2 L61 to L62: "Another problem that will be involved is the selection of computational spatial resolution." Suggest making this more concise: "Another problem is the selection of…".

**Response and change:** Thank you for the comment. The change has been done. (P2 L64)

**Comment:** P3 L66: "squared meters" to "square meters" or "meters squared".

**Response and change:** The correction has been done. (P3 L69)

**Comment:** P4 L102 to L103: If URLs are placed in the text, please add date of access.

**Response and change:** The date of access has been added. The first URL (http://hydro.ou.edu) was assessed on 23 December 2014. The second URL (www.servir.net) was assessed on 15 September 2016. (P4 L105)

**Comment:** P9 L237 to L239: The paragraph mentions that TWI is related to soil moisture through the wetness coefficient, and the reader is directed to another paper. Please briefly explain this relationship to the reader in this section, just enough for them to understand the basics.

**Response and change:** Thank you for the comment. The relationship between $K_w$ and $TWI$ at the coarse resolution ($K_{w,coarse}$ and $TWI_{coarse}$) is first detected, and the concave and convex areas are also distinguished. Then this relation is used to calculate $K_w$ and TWI at the fine resolution ($K_{w,fine}$ and $TWI_{fine}$), which is further used to fix the soil moisture at fine resolution. We have added this explanation in the revised version. (P9 L242)

**Comment:** P19 L446: Change "… the model behaves sensitive…" to "… the model is sensitive…".

**Response and change:** The change has been done. (P20 L453)

**Comment:** P28 L605: Change "… containing two resolutions datasets…" to "… containing two datasets with different resolutions…"

**Response and change:** The change has been done. (P28 611)

**Comment:** P28 L605 to L606: "For the purpose that the model is practicably performed in the case of large scale, we parallelized the program for efficiency." Suggest rewording this sentence for clarity, particularly the first part.

**Response and change:** We have improved it for clarity. (P28 L613)

**Comment:** P28 L609: Change "literatures" to "literature."

**Response and change:** The change has been done. (P28 L616)

**Comment:** P28 L614: Remove "as a matter of fact".

**Response and change:** We have removed it. (P29 621)

**Comment:** P28 L616: Remove "we want to point out".

**Response and change:** We have removed it. (P29 L623)

**Comment:** P30 L670: There seems to be a strange tab/space here in the manuscript, maybe "In summary…" was supposed to be part of a new paragraph?

**Response and change:** Thank you for the check. The text starting from the "In summary" is a new paragraph and we have corrected it. (P30 L678)

We finally went through the text again and fixed several typos (tense, grammar, and spelling). See P4 L98, P12 L301, P14 L361, P26 L570, and P26 L572.